# Deep Active Learning by Leveraging Training Dynamics

**Haonan Wang**[1], **Wei Huang**[2], **Ziwei Wu**[1], **Andrew Margenot**[1],
**Hanghang Tong**[1], **Jingrui He**[1]

[1]University of Illinois Urbana-Champaign
[2]University of New South Wales
[1]*{haonan3,ziweiwu2,margenot,htong,jingrui}@illinois.edu*
[2]*{weihuang.uts}@gmail.com*

## Abstract

Active learning theories and methods have been extensively studied in classical statistical learning settings. However, deep active learning, i.e., active learning with deep learning models, is usually based on empirical criteria without solid theoretical justification, thus suffering from heavy doubts when some of those fail to provide benefits in real applications. In this paper, by exploring the connection between the generalization performance and the training dynamics, we propose a theory-driven deep active learning method (***dynamicAL***) which selects samples to maximize training dynamics. In particular, we prove that the convergence speed of training and the generalization performance are positively correlated under the ultra-wide condition and show that maximizing the training dynamics leads to better generalization performance. Furthermore, to scale up to large deep neural networks and data sets, we introduce two relaxations for the subset selection problem and reduce the time complexity from polynomial to constant. Empirical results show that *dynamicAL* not only outperforms the other baselines consistently but also scales well on large deep learning models. We hope our work would inspire more attempts on bridging the theoretical findings of deep networks and practical impacts of deep active learning in real applications.

## 1 Introduction

Training deep learning (DL) models usually requires large amount of high-quality labeled data [1] to optimize a model with a massive number of parameters. The acquisition of such annotated data is usually time-consuming and expensive, making it unaffordable in the fields that require high domain expertise. A promising approach for minimizing the labeling effort is active learning (AL), which aims to identify and label the maximally informative samples, so that a high-performing classifier can be trained with minimal labeling effort [2]. Under classical statistical learning settings, theories of active learning have been extensively studied from the perspective of VC dimension [3]. As a result, a variety of methods have been proposed, such as (i) the version-space-based approaches, which require maintaining a set of models [4, 5], and (ii) the clustering-based approaches, which assume that the data within the same cluster have pure labels [6].

However, the theoretical analyses for these classical settings may not hold for over-parameterized deep neural networks where the traditional wisdom is ineffective [1]. For example, margin-based methods select the labeling examples in the vicinity of the learned decision boundary [7, 8]. However, in the over-parameterized regime, every labeled example could potentially be near the learned decision boundary [9]. As a result, theoretically, such analysis can hardly guide us to design practical active

36th Conference on Neural Information Processing Systems (NeurIPS 2022).

learning methods. Besides, empirically, multiple deep active learning works, borrowing observations and insights from the classical theories and methods, have been observed unable to outperform their passive learning counterparts in a few application scenarios [10, 11].

On the other hand, the analysis of neural network's optimization and generalization performance has witnessed several exciting developments in recent years in terms of the deep learning theory [12, 13, 14]. It is shown that the training dynamics of deep neural networks using gradient descent can be characterized by the Neural Tangent Kernel (NTK) of infinite [12] or finite [15] width networks. This is further leveraged to characterize the generalization of over-parameterized networks through Rademacher complexity analysis [13, 16]. We are therefore inspired to ask: How can we design a practical and generic active learning method for deep neural networks with theoretical justifications?

To answer this question, we firstly explore the connection between the model performance on testing data and the convergence speed on training data for the over-parameterized deep neural networks. Based on the NTK framework [12, 13], we theoretically show that if a deep neural network converges faster ("Train Faster"), then it tends to have better generalization performance ("Generalize Better"), which matches the existing observations [17, 18, 19, 20, 21]. Motivated by the aforementioned connection, we first introduce *Training Dynamics*, the derivative of training loss with respect to iteration, as a proxy to quantitatively describe the training process. On top of it, we formally propose our generic and theoretically-motivated deep active learning method, *dynamicAL*, which will query labels for a subset of unlabeled samples that maximally increase the training dynamics. In order to compute the training dynamics by merely using the unlabeled samples, we leverage two relaxations *Pseudo-labeling* and *Subset Approximation* to solve this non-trivial subset selection problem. Our relaxed approaches are capable of effectively estimating the training dynamics as well as efficiently solving the subset selection problem by reducing the complexity from $O(N^b)$ to $O(b)$.

In theory, we coin a new term *Alignment* to measure the length of the label vector's projection on the neural tangent kernel space. Then, we demonstrate that higher alignment usually comes with a faster convergence speed and a lower generalization bound. Furthermore, with the help of the maximum mean discrepancy [22], we extend the previous analysis to an active learning setting where the i.i.d. assumption may not hold. Finally, we show that alignment is positively correlated with our active learning goal, training dynamics, which implies that maximizing training dynamics will lead to better generalization performance.

Regarding experiments, we have empirically verified our theory by conducting extensive experiments on three datasets, CIFAR10 [23], SVHN [24], and Caltech101 [25] using three types of network structures: vanilla CNN, ResNet [26], and VGG [27]. We first show that the result of the subset selection problem delivered by the subset approximation is close to the global optimal solution. Furthermore, under the active learning setting, our method not only outperforms other baselines but also scales well on large deep learning models.

The main contributions of our paper can be summarized as follows:

- We propose a theory-driven deep active learning method, *dynamicAL*, inspired by the observation of "train faster, generalize better". To this end, we introduce the Training Dynamics, as a proxy to describe the training process.

- We demonstrate that the convergence speed of training and the generalization performance is strongly (positively) correlated under the ultra-wide condition; we also show that maximizing the training dynamics will lead to a lower generalization error in the scenario of active learning.

- Our method is easy to implement. We conduct extensive experiments to evaluate the effectiveness of *dynamicAL* and empirically show that our method consistently outperforms other methods in a wide range of active learning settings.

## 2  Background

**Notation.** We use the random variable $x \in \mathcal{X}$ to represent the input data feature and $y \in \mathcal{Y}$ as the label where $K$ is the number of classes and $[K] := \{1, 2, ..., K\}$. We are given non-degenerated a data source $D$ with unknown distribution $p(x, y)$. We further denote the concatenation of $x$ as $X = [x_1, x_2, ..., x_M]^\top$ and that of $y$ as $Y = [y_1, y_2, ..., y_M]^\top$. We consider a deep learning classifier $h_\theta(x) = \text{argmax } \sigma(f(x; \theta)) : x \to y$ parameterized by $\theta \in \mathbb{R}^p$, where $\sigma(\cdot)$ is the softmax function and $f$ is a neural network. Let $\otimes$ be the Kronecker Product and $I_K \in \mathbb{R}^{K \times K}$ be an identity matrix.

**Active learning.** The goal of active learning is to improve the learning efficiency of a model with a limited labeling budget. In this work, we consider the pool-based AL setup, where a finite data set $S = \{(x_l, y_l)\}_{l=1}^{M}$ with $M$ points are $i.i.d.$ sampled from $p(x, y)$ as the (initial) labeled set. The AL model receives an unlabeled data set $U$ sampled from $p(x)$ and request labels according to $p(y|x)$ for any $x \in U$ in each query round. There are $R$ rounds in total, and for each round, a query set $Q$ consisting of $b$ unlabeled samples can be queried. The total budget size $B = b \times R$.

**Neural Tangent Kernel.** The Neural Tangent Kernel [12] has been widely applied to analyze the dynamics of neural networks. If a neural network is sufficiently wide, properly initialized, and trained by gradient descent with infinitesimal step size (*i.e.*, gradient flow), then the neural network is equivalent to kernel regression predictor with a deterministic kernel $\Theta(\cdot, \cdot)$, called Neural Tangent Kernel (NTK). When minimizing the mean squared error loss, at the iteration $t$, the dynamics of the neural network $f$ has a closed-form expression:

$$\frac{df(\mathcal{X}; \theta(t))}{dt} = -\mathcal{K}_t(\mathcal{X}, \mathcal{X})\left(f(\mathcal{X}; \theta(t)) - \mathcal{Y}\right), \tag{1}$$

where $\theta(t)$ denotes the parameter of the neural network at iteration $t$, $\mathcal{K}_t(\mathcal{X}, \mathcal{X}) \in \mathbb{R}^{|\mathcal{X}| \times K \times |\mathcal{X}| \times K}$ is called the empirical NTK and $\mathcal{K}_t^{i,j}(x, x') = \nabla_\theta f^i(x; \theta(t))^\top \nabla_\theta f^j(x'; \theta(t))$ is the inner product of the gradient of the $i$-th class probability and the gradient of the $j$-th class probability for two samples $x, x' \in \mathcal{X}$ and $i, j \in [K]$. The time-variant kernel $\mathcal{K}_t(\cdot, \cdot)$ is equivalent to the (time-invariant) NTK with a high probability, i.e., if the neural network is sufficiently wide and properly initialized, then:

$$\mathcal{K}_t(\mathcal{X}, \mathcal{X}) = \Theta(\mathcal{X}, \mathcal{X}) \otimes I_K. \tag{2}$$

The final learned neural network at iteration $t$, is equivalent to the kernel regression solution with respect to the NTK [14]. For any input $x$ and training data $\{X, Y\}$ we have,

$$f(x; \theta(t)) \approx \Theta(x, X)^\top \Theta(X, X)^{-1}(I - e^{-\eta \Theta(X, X)t})Y, \tag{3}$$

where $\eta$ is the learning rate, $\Theta(x, X)$ is the NTK matrix between input $x$ and all samples in training data $X$.

## 3 Method

In section 3.1, we introduce the notion of training dynamics which can be used to describe the training process. Then, in section 3.2, based on the training dynamics, we propose *dynamicAL*. In section 3.3, we discuss the connection between *dynamicAL* and existing deep active learning methods.

### 3.1 Training dynamics

In this section, we introduce the notion of training dynamics. The cross-entropy loss over the labeled set $S$ is defined as:

$$
\begin{aligned}
L(S) &= \sum_{(x_l, y_l) \in S} \ell(f(x_l; \theta), y_l) \\
&= -\sum_{(x_l, y_l) \in S} \sum_{i \in [K]} y_l^i \log \sigma^i(f(x_l; \theta)),
\end{aligned} \tag{4}
$$

where $\sigma^i(f(x; \theta)) = \frac{\exp(f^i(x; \theta))}{\sum_j \exp(f^j(x; \theta))}$. We first analyze the dynamics of the training loss, with respect to iteration $t$, on one labeled sample (derivation is in Appendix A.1):

$$\frac{\partial \ell(f(x; \theta), y)}{\partial t} = -\sum_i \left(y^i - \sigma^i(f(x; \theta))\right) \nabla_\theta f^i(x; \theta) \nabla_t^\top \theta. \tag{5}$$

For neural networks trained by gradient descent, if the learning rate $\eta$ is small, then $\nabla_t \theta = \theta_{t+1} - \theta_t = -\eta \frac{\partial \sum_{(x_l, y_l) \in S} \ell(f(x_l; \theta), y_l)}{\partial \theta}$. Taking the partial derivative of the training loss with respect to the parameters, we have (the derivation of the following equation can be found in Appendix A.2):

$$\frac{\partial \ell(f(x; \theta), y)}{\partial \theta} = \sum_{j \in [K]} \left(\sigma^j(f(x; \theta)) - y^j\right) \frac{\partial f^j(x; \theta)}{\partial \theta}. \tag{6}$$

Therefore, we can further get the following result for the dynamics of training loss:

$$\frac{\partial \ell(f(x;\theta),y)}{\partial t} = -\eta \sum_i \left(\sigma^i(f(x;\theta)) - y^i\right) \sum_j \sum_{(x_{l'},y_{l'})\in S} \nabla_\theta f^i(x;\theta)^\top \nabla_\theta f^j(x_{l'};\theta)\left(\sigma^j(f(x_{l'};\theta)) - y^j_{l'}\right).$$
(7)

Furthermore, we define $d^i(X,Y) = \sigma^i(f(X;\theta)) - Y^i$ and $Y^i$ is the label vector of all samples for $i$-th class. Then, the *training dynamics* (dynamics of training loss) over training set $S$, computed with the empirical NTK $\mathcal{K}^{ij}(X,X)$, is denoted by $G(S) \in \mathbb{R}$:

$$G(S) = -\frac{1}{\eta} \sum_{(x_l,y_l)\in S} \frac{\partial \ell(f(x_l;\theta),y_l)}{\partial t} = \sum_i \sum_j d^i(X,Y)^\top \mathcal{K}^{ij}(X,X)d^j(X,Y).$$
(8)

## 3.2 Active learning by activating training dynamics

Before we present *dynamicAL*, we state Proposition 1, which serves as the theoretical guidance for *dynamicAL* and will be proved in Section 4.

**Proposition 1.** *For deep neural networks, converging faster leads to a lower worst-case generalization error.*

Motivated by the connection between convergence speed and generalization performance, we propose the general-purpose active learning method, *dynamicAL*, which aims to accelerate the convergence by querying labels for unlabeled samples. As we described in the previous section, the training dynamics can be used to describe the training process. Therefore, we employ the training dynamics as a proxy to design an active learning method. Specifically, at each query round, *dynamicAL* will query labels for samples which maximize the training dynamics $G(S)$, *i.e.*,

$$Q = \text{argmax}_{Q\subseteq U} G(S \cup \overline{Q}), \ s.t. \ |Q| = b,$$
(9)

where $\overline{Q}$ is the corresponding data set for $Q$ with ground-truth labels. Notice that when applying the above objective in practice, we are facing two major challenges. First, $G(S \cup \overline{Q})$ cannot be directly computed, because the label information of unlabeled examples is not available before the query. Second, the subset selection problem can be computationally prohibitive if enumerating all possible sets with size $b$. Therefore, we employ the following two relaxations to make this maximization problem to be solved with constant time complexity.

**Pseudo labeling.** To estimate the training dynamics, we use the predicted label $\hat{y}_u$ for sample $x_u$ in the unlabeled data set $U$ to compute $G$. Note, the effectiveness of this adaptation has been demonstrated in the recent gradient-based methods [11, 28], which compute the gradient as if the model's current prediction on the example is the true label. Therefore, the maximization problem in Equation (9) is changed to,

$$Q = \text{argmax}_{Q\subseteq U} G(S \cup \widehat{Q}).$$
(10)

where $\widehat{Q}$ is the corresponding data set for $Q$ with pseudo labels $\widehat{Y}_Q$.

**Subset approximation.** The subset selection problem of Equation (10) still requires enumerating all possible subsets of $U$ with size $b$, which is $O(n^b)$. We simplify the selection problem to the following problem without causing any change on the result,

$$\text{argmax}_{Q\subseteq U} G(S \cup \widehat{Q}) = \text{argmax}_{Q\subseteq U} \Delta(\widehat{Q}|S),$$
(11)

where $\Delta(\widehat{Q}|S) = G(S \cup \widehat{Q}) - G(S)$ is defined as the change of training dynamics. We approximate the change of training dynamics caused by query set $Q$ using the summation of the change of training dynamics caused by each sample in the query set. Then the maximization problem can be converted to Equation (12) which can be solved by a greedy algorithm with $O(b)$.

$$Q = \text{argmax}_{Q\subseteq U} \sum_{(x,\widehat{y})\in \widehat{Q}} \Delta(\{(x,\widehat{y})\}|S), \ s.t. \ |Q| = b.$$
(12)

To further show the approximated result is reasonably good, we decompose the change of training dynamics as (derivation in Appendix A.4):

$$\Delta(\widehat{Q}|S) = \sum_{(x,\widehat{y})\in \widehat{Q}} \Delta(\{(x,\widehat{y})\}|S) + \sum_{(x,\hat{y}),(x',\hat{y}')\in \widehat{Q}} d^i(x,\hat{y})^\top \mathcal{K}^{ij}(x,x')d^j(x',\hat{y}'),$$
(13)

where $\mathcal{K}^{ij}(x, x')$ is the empirical NTK. The first term in the right hand side is the approximated change of training dynamics. Then, we further define the *Approximation Ratio* (14) which measures the approximation quality,

$$R(\widehat{Q}|S) = \frac{\sum_{(x,\widehat{y})\in\widehat{Q}} \Delta(\{(x,\widehat{y})\}|S)}{\Delta(\widehat{Q}|S)}. \tag{14}$$

We empirically measure the expectation of the Approximation Ratio on two data sets with two different neural networks under three different batch sizes. As shown in Figure 4, the expectation $\mathbb{E}_{Q\sim U} R(\widehat{Q}|S) \approx 1$ when the model is converged. Therefore, the approximated result delivered by the greedy algorithm is close to the global optimal solution of the original maximization problem, Equation (10), especially when the model is converged.

Based on the above two approximations, we present the proposed method *dynamicAL* in Algorithm 1. As described below, the algorithm starts by training a neural network $f(\cdot;\theta)$ on the initial labeled set $S$ until convergence. Then, for every unlabeled sample $x_u$, we compute pseudo label $\widehat{y}_u$ and the change of training dynamics $\Delta(\{(x_u,\widehat{y}_u)\}|S)$. After that, *dynamicAL* will query labels for top-$b$ samples causing the maximal change on training dynamics, train the neural network on the extended labeled set, and repeat the process. Note, to keep close to the theoretical analysis, re-initialization is not used after each query, which also enables *dynamicAL* to get rid of the computational overhead of retraining the deep neural networks every time.

---

**Algorithm 1** Deep Active Learning by Leveraging Training Dynamics

---

**Input:** Neural network $f(\cdot;\theta)$, unlabeled sample set $U$, initial labeled set $S$, number of query round $R$, query batch size $b$.
**for** $r = 1$ **to** $R$ **do**
    Train $f(\cdot;\theta)$ on $S$ with cross-entropy loss until convergence.
    **for** $x_u \in U$ **do**
        Compute its pseudo label $\widehat{y}_u = \text{argmax} f(x_u;\theta)$.
        Compute $\Delta(\{(x_u,\widehat{y}_u)\}|S)$.
    **end for**
    Select $b$ query samples $Q$ with the highest $\Delta$ values, and request their labels from the oracle.
    Update the labeled data set $S = S \cup \overline{Q}$ .
**end for**
**return** Final model $f(\cdot;\theta)$.

---

### 3.3 Relation to existing methods

Although existing deep active learning methods are usually designed based on heuristic criteria, some of them have empirically shown their effectiveness [11, 29, 30]. We surprisingly found that our theoretically-motivated method *dynamicAL* has some connections with those existing methods from the perspective of active learning criterion. The proposed active learning criterion in Equation (12) can be explicitly written as (derivation in Appendix A.5):

$$\Delta(\{(x_u,\widehat{y}_u)\}|S) = \|\nabla_\theta \ell(f(x_u;\theta), \widehat{y}_u)\|^2 + 2 \sum_{(x,y)\in S} \nabla_\theta \ell(f(x_u;\theta), \widehat{y}_u)^\top \nabla_\theta \ell(f(x;\theta), y). \tag{15}$$

**Note.** The first term of the right-hand side can be interpreted as the square of gradient length (2-norm) which reflects the uncertainty of the model on the example and has been wildly used as an active learning criterion in some existing works [30, 11, 31]. The second term can be viewed as the influence function [32] with identity hessian matrix. And recently, [29] has empirically shown that the effectiveness of using the influence function with identity hessian matrix as active learning criterion. We hope our theoretical analysis can also shed some light on the interpretation of previous methods.

## 4 Theoretical analysis

In this section, we study the correlation between the convergence rate of the training loss and the generalization error under the ultra-wide condition [12, 13]. We define a measure named *alignment*

to quantify the convergence rate and further show its connection with generalization bound. The analysis provides a theoretical guarantee for the phenomenon of "Train Faster, Generalize Better" as well as our active learning method *dynamicAL* with a rigorous treatment. Finally, we show that the active learning proxy, training dynamics, is correlated with alignment, which indicates that increasing the training dynamics leads to larger convergence rate and better generalization performance. We leave all proofs of theorems and details of verification experiments in Appendix B and D respectively.

## 4.1 Train faster provably generalize better

Given an ultra-wide neural network, the gradient descent can achieve a near-zero training error [12, 33] and its generalization ability in unseen data can be bounded [13]. It is shown that both the convergence and generalization of a neural network can be analyzed using the NTK [13]. However, the question what is the relation between the convergence rate and the generalization bound has not been answered. We formally give a solution by introducing the concept of *alignment*, which is defined as follows:

**Definition 1** (Alignment). *Given a data set $S = \{X, Y\}$, the alignment is a measure of correlation between $X$ and $Y$ projected in the NTK space. In particular, the alignment can be computed by $\mathcal{A}(X, Y) = \mathrm{Tr}[Y^\top \boldsymbol{\Theta}(X, X) Y] = \sum_{k=1}^{K} \sum_{i=1}^{n} \lambda_i (\vec{v}_i^\top Y^k)^2$.*

In the following, we will demonstrate why "Train Faster" leads to "Generalize Better" through alignment. In particular, the relation of the convergence rate and the generalization bound with alignment is analyzed. The convergence rate of gradient descent for ultra-wide networks is presented in following lemma:

**Lemma 1** (Convergence Analysis with NTK, Theorem 4.1 of [13]). *Suppose $\lambda_0 = \lambda_{\min}(\boldsymbol{\Theta}) > 0$ for all subsets of data samples. For $\delta \in (0, 1)$, if $m = \Omega(\frac{n^7}{\lambda_0^4 \delta^4 \epsilon^2})$ and $\eta = O(\frac{\lambda_0}{n^2})$, with probability at least $1 - \delta$, the network can achieve near-zero training error,*

$$\|Y - f(X; \theta(t))\|_2 = \sqrt{\sum_{k=1}^{K} \sum_{i=1}^{n} (1 - \eta \lambda_i)^{2t} (\vec{v}_i^\top Y^k)^2} \pm \epsilon, \tag{16}$$

*where $n$ denotes the number of training samples and $m$ denotes the width of hidden layers. The NTK $\boldsymbol{\Theta} = V^\top \Lambda V$ with $\Lambda = \{\lambda_i\}_{i=1}^{n}$ is a diagonal matrix of eigenvalues and $V = \{\vec{v}_i\}_{i=1}^{n}$ is a unitary matrix.*

In this lemma, we take mean square error (MSE) loss as an example for the convenience of illustration. The conclusion can be extended to other loss functions such as cross-entropy loss (see Appendix B.2 in [14]). From the lemma, we find the convergence rate is governed by the dominant term (16) as $\mathcal{E}_t(X, Y) = \sqrt{\sum_{k=1}^{K} \sum_{i=1}^{n} (1 - \eta \lambda_i)^{2t} (\vec{v}_i^\top Y^k)^2}$, which is correlated with the *alignment*:

**Theorem 1** (Relationship between the convergence rate and alignment). *Under the same assumptions as in Lemma 1, the convergence rate described by $\mathcal{E}_t$ satisfies,*

$$\mathrm{Tr}[Y^\top Y] - 2t\eta \mathcal{A}(X, Y) \leq \mathcal{E}_t^2(X, Y) \leq \mathrm{Tr}[Y^\top Y] - \eta \mathcal{A}(X, Y). \tag{17}$$

**Remark 1.** *In the above theorem, we demonstrate that the alignment can measure the convergence rate. Especially, we find that both the upper bound and the lower bound of error $\mathcal{E}_t(X, Y)$ are inversely proportional to the alignment, which implies that higher alignment will lead to achieving faster convergence.*

Now we analyze the generalization performance of the proposed method through complexity analysis. We demonstrate that the ultra-wide networks can achieve a reasonable generalization bound.

**Lemma 2** (Generalization bound with NTK, Theorem 5.1 of [13]). *Suppose data $S = \{(x_i, y_i)\}_{i=1}^{n}$ are i.i.d. samples from a non-degenerate distribution $p(x, y)$, and $m \geq \mathrm{poly}(n, \lambda_0^{-1}, \delta^{-1})$. Consider any loss function $\ell : \mathbb{R} \times \mathbb{R} \to [0, 1]$ that is 1-Lipschitz, then with probability at least $1 - \delta$ over the random initialization, the network trained by gradient descent for $T \geq \Omega(\frac{1}{\eta \lambda_0} \log \frac{n}{\delta})$ iterations has population risk $\mathcal{L}_p = \mathbb{E}_{(x,y) \sim p(x,y)}[\ell(f_T(x; \theta), y)]$ that is bounded as follows:*

$$\mathcal{L}_p \leq \sqrt{\frac{2 \mathrm{Tr}[Y^\top \boldsymbol{\Theta}^{-1}(X, X) Y]}{n}} + O\left(\sqrt{\frac{\log \frac{n}{\lambda_0 \delta}}{n}}\right). \tag{18}$$

In this lemma, we show that the dominant term in the generalization upper bound is $\mathcal{B}(X, Y) = \sqrt{\frac{2 \operatorname{Tr}[Y^\top \boldsymbol{\Theta}^{-1} Y]}{n}}$. In the following theorem, we further prove that this bound is inversely proportional to the alignment $\mathcal{A}(X, Y)$.

**Theorem 2** (Relationship between the generalization bound and alignment). *Under the same assumptions as in Lemma 2, if we define the generalization upper bound as $\mathcal{B}(X, Y) = \sqrt{\frac{2 \operatorname{Tr}[Y^\top \boldsymbol{\Theta}^{-1} Y]}{n}}$, then it can be bounded with the alignment as follows:*

$$\frac{\operatorname{Tr}^2[Y^\top Y]}{\mathcal{A}(X, Y)} \le \frac{n}{2} \mathcal{B}^2(X, Y) \le \frac{\lambda_{max}}{\lambda_{min}} \frac{\operatorname{Tr}^2[Y^\top Y]}{\mathcal{A}(X, Y)}. \tag{19}$$

**Remark 2.** *Theorems 1 and 2 reveal that the cause for the correlated phenomenons "Train Faster" and "Generalize Better" is the projection of label vector on the NTK space (alignment).*

## 4.2 " Train Faster, Generalize Better " for active learning

In the NTK framework [13], the empirical average requires data in $S$ is *i.i.d.* samples (Lemma 2). However, this assumption may not hold in the active learning setting with multiple query rounds, because the training data is composed by *i.i.d.* sampled initial label set and samples queried by active learning policy. To extend the previous analysis principle to active learning, we follow [34] to reformulate the Lemma 2 as:

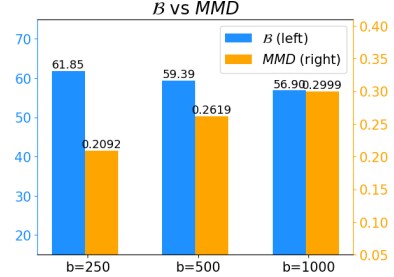

Figure 1: Comparison between Empirical Generalization Bound and MMD.

$$\mathcal{L}_p \le (\mathcal{L}_p - \mathcal{L}_q) + \sqrt{\frac{2 \operatorname{Tr}[Y^\top \boldsymbol{\Theta}^{-1}(X, X) Y]}{n}} + O\left( \sqrt{\frac{\log \frac{n}{\lambda_0 \delta}}{n}} \right), \tag{20}$$

where $\mathcal{L}_q = \mathbb{E}_{(x,y) \sim q(x,y)}[\ell(f(x; \theta), y)]$, $q(x, y)$ denotes the data distribution after query, and $X, Y$ includes initial training samples and samples after query. There is a new term in the upper bound, which is the difference between the true risk under different data distributions.

$$\mathcal{L}_p - \mathcal{L}_q = \mathbb{E}_{(x,y) \sim p(x,y)}[\ell(f(x; \theta), y)] - \mathbb{E}_{(x,y) \sim q(x,y)}[\ell(f(x; \theta), y)] \tag{21}$$

Though in active learning the data distribution for the labeled samples may be different from the original distribution, they share the same conditional probability $p(y|x)$. We define $g(x) = \int_y \ell(f(x; \theta), y) p(y|x) dy$, and then we have:

$$\mathcal{L}_p - \mathcal{L}_q = \int_x g(x) p(x) dx - \int_x g(x) q(x) dx. \tag{22}$$

To measure the distance between two distributions, we employ the Maximum Mean Discrepancy (MMD) with neural tangent kernel [35] (derivation in Appendix B.3).

$$\mathcal{L}_p - \mathcal{L}_q \le \operatorname{MMD}(S_0, S, \mathcal{H}_{\boldsymbol{\Theta}}) + O\left( \sqrt{\frac{C \ln(1/\delta)}{n}} \right). \tag{23}$$

Slightly overloading the notation, we denote the initial labeled set as $S_0$, $\mathcal{H}_{\boldsymbol{\Theta}}$ as the associated Reproducing Kernel Hilbert Space for the NTK $\boldsymbol{\Theta}$, and $\forall x, x' \in S$, $\boldsymbol{\Theta}(x, x') \le C$. Note, $\operatorname{MMD}(S_0, S, \mathcal{H}_{\boldsymbol{\Theta}})$ is the empirical measure for $\operatorname{MMD}(p(x), q(x), \mathcal{H}_{\boldsymbol{\Theta}})$. We empirically compute MMD and the dominant term of the generalization upper bound $\mathcal{B}$ under the active learning setting with our method *dynamicAL*. As shown in Figure 1, on CIFAR10 with a CNN target model (three convolutional layers with global average pooling), the initial labeled set size $|S| = 500$, query round $R = 1$ and budget size $b \in \{250, 500, 1000\}$, we observe that, under different active learning settings, the MMD is always much smaller than the $\mathcal{B}$. Besides, we further investigate the MMD and $\mathcal{B}$ for $R \ge 2$ and observe the similar results. Therefore, the lemma 2 still holds for the target model with *dynamicAL*. More results and discussions for $R \ge 2$ are in Appendix E.4 and the computation details of MMD and NTK are in Appendix D.1.

### 4.3 Alignment and training dynamics in active learning

In this section, we show the relationship between the alignment and the training dynamics. To be consistent with the previous theoretical analysis (Theorem 1 and 2), we use the training dynamics with mean square error under the ultra-width condition, which can be expressed as $G_{MSE}(S) = \text{Tr}\left[(f(X;\theta) - Y)^{\top}\Theta(X,X)(f(X;\theta) - Y)\right]$. Due to the limited space, we leave the derivation in Appendix A.3. To further quantitatively evaluate the correlation between $G_{MSE}(S \cup \overline{Q})$ and $\mathcal{A}(X\|X_Q, Y\|Y_Q)$, we utilize the Kendall $\tau$ coefficient [36] to empirically measure their relation. As shown in Figure 2, for CNN on CIFAR10 with active learning setting, where $|S| = 500$ and $|\overline{Q}| = 250$, there is a strong agreement between $G_{MSE}(S \cup \overline{Q})$ and $\mathcal{A}(X\|X_Q, Y\|Y_Q)$, which further indicates that increasing the training dynamics will lead to a faster convergence and better generalization performance. More details about this verification experiment are in Appendix D.2.

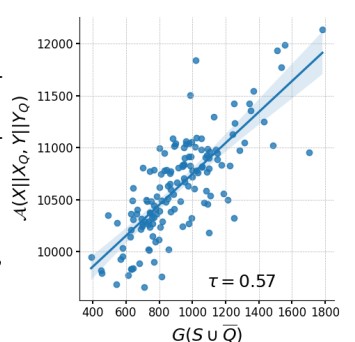

Figure 2: Relation between Alignment and Training Dynamics.

## 5 Experiments

### 5.1 Experiment setup

**Baselines.** We compare *dynamicAL* with the following eight baselines: Random, Corset, Confidence Sampling (Conf), Margin Sampling (Marg), Entropy, and Active Learning by Learning (ALBL), Batch Active learning by Diverse Gradient Embeddings (BADGE). Description of baseline methods is in Appendix E.1.

**Data sets and Target Model.** We evaluate all the methods on three benchmark data sets, namely, CIFAR10 [23], SVHN [24], and Caltech101 [25]. We use accuracy as the evaluation metric and report the mean value of 5 runs. We consider three neural network architectures: vanilla CNN, ResNet18 [26], and VGG11 [27]. For each model, we keep the hyper-parameters used in their official implementations. More information about the implementation is in Appendix C.1.

**Active Learning Protocol.** Following the previous evaluation protocol [11], we compare all those active learning methods in a batch-mode setup with an initial set size $M = 500$ for all those three data sets, batch size $b$ varying from $\{250, 500, 1000\}$. For the selection of test set, we use the benchmark split of the CIFAR10 [23], SVHN [24] and sample 20% from each class to form the test set for the Caltech101 [25].

### 5.2 Results and analysis

The main experimental results have been provided as plots due to the limited space. We also provide tables in which we report the mean and standard deviation for each plot in Appendix E.3.

**Overall results.** The average test accuracy at each query round is shown in Figure 3. Our method *dynamicAL* can consistently outperform other methods for all query rounds. This suggests that *dynamicAL* is a good choice regardless of the labeling budget. And, we notice *dynamicAL* can work well on data sets with a large class number, such as Caltech101. However, the previous state-of-the-art method, BADGE, cannot be scaled up to those data sets, because the required memory is linear with the number of classes. Besides, because *dynamicAL* depends on pseudo labeling, a relatively large initial labeled set can provide advantages for *dynamicAL*. Therefore, it is important to examine whether *dynamicAL* can work well with a small initial labeled set. As shown in Figure 3, *dynamicAL* is able to work well with a relatively small initial labeled set ($M = 500$). Due to the limited space, we only show the result under three different settings in Figure 3. More evaluation results are in Appendix E.2. Moreover, although the re-initialization trick makes *dynamicAL* deviate from the dynamics analysis, we investigate the effect of it to *dynamicAL* and provide the empirical observations and analysis in Appendix E.5.

**Effect of query size and query round.** Given the total label budget $B$, the increasing of query size always leads to the decreasing of query round. We study the influence of different query size

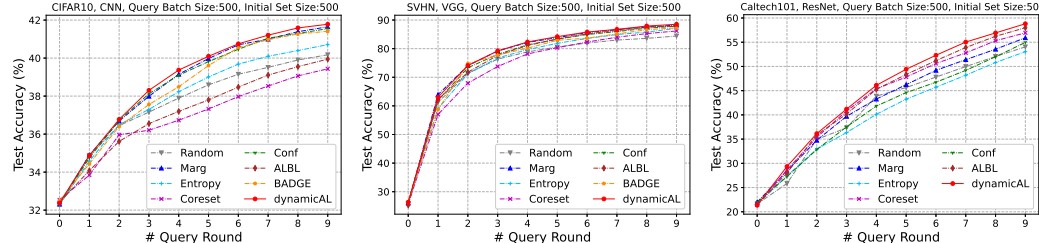

Figure 3: Active learning test accuracy versus the number of query rounds for a range of conditions.

and query round on *dynamicAL* from two perspectives. First, we study the expected approximation ratio with different query batch sizes on different data sets. As shown in Figure 4, under different settings the expected approximation ratio always converges to 1 with the increase of training epochs, which further indicates that the query set selected by using the approximated change of training dynamics is a reasonably good result for the query set selection problem. Second, we study influence of query round for actual performance of target models. The performance for different target models on different data sets with total budge size $B = 1000$ is shown in Table 1. For certain query budget, our active learning algorithm can be further improved if more query rounds are allowed.

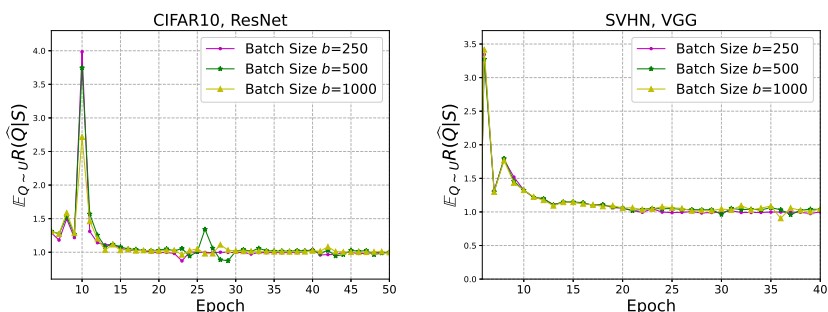

Figure 4: The Expectation of the Approximation Ratio with different query batch sizes $b$.

Table 1: Accuracy of *dynamicAL* with different query batch size $b$.

| Setting | CIFAR10+CNN | CIFAR10+Resnet | SVHN+VGG | Caltech101+Resnet |
|---|---|---|---|---|
| $R = 10, b = 100$ | **36.84** | **40.92** | **76.34** | **37.06** |
| $R = 4, b = 250$ | 36.72 | 40.78 | 75.26 | 36.48 |
| $R = 2, b = 500$ | 36.71 | 40.46 | 74.10 | 35.91 |
| $R = 1, b = 1000$ | 36.67 | 40.09 | 70.04 | 33.82 |

**Comparison with different variants.** The active learning criterion of *dynamicAL* can be written as $\sum_{(x,y)\in S} \|\nabla_\theta \ell(f(x;\theta_u), \hat{y}_u)\|^2 + \gamma \nabla_\theta \ell(f(x_u;\theta), \hat{y}_u)^\top \nabla_\theta \ell(f(x;\theta), y)$. We empirically show the performance for $\gamma \in \{0, 1, 2, \infty\}$ in Figure 5. With $\gamma = 0$, the criterion is close to the expected gradient length method [31]. And with $\gamma = \infty$, the selected samples are same with the samples selected by using the influence function with identity hessian matrix criterion [29]. As shown in Figure 5, the model achieves the best performance with $\gamma = 2$, which is aligned with the value indicated by the theoretical analysis (Equation 15). The result confirms the importance of theoretical analysis for the design of deep active learning methods.

## 6 Related work

**Neural Tangent Kernel (NTK):** Recent study has shown that under proper conditions, an infinite-width neural network can be simplified as a linear model with Neural Tangent Kernel (NTK) [12]. Since then, NTK has become a powerful theoretical tool to analyze the behavior of deep learning architecture (CNN, GNN, RNN) [33, 37, 38], random initialization [39], stochastic neural network [40], and graph neural network [41] from its output dynamics and to characterize the convergence and generalization error [13]. Besides, [15] studies the finite-width NTK, aiming at making the NTK more practical.

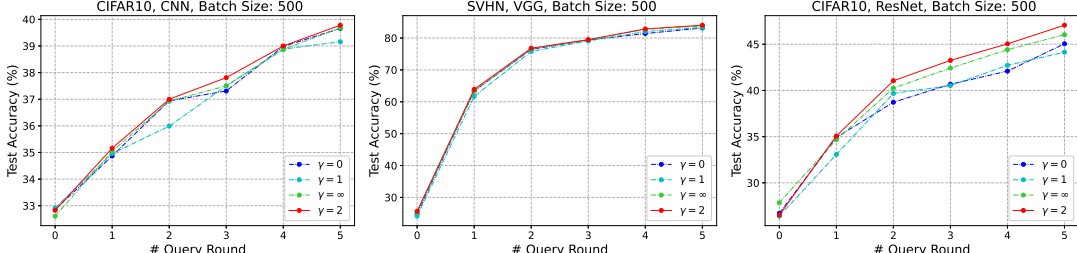

Figure 5: Test Accuracy of different variants.

**Active Learning:** Active learning aims at interactively query labels for unlabeled data points to maximize model performances [2]. Among others, there are two popular strategies for active learning, *i.e.*, diversity sampling [42, 43, 44] and uncertainty sampling [45, 46, 47, 11, 48, 49, 29]. Recently, several papers proposed to use gradient to measure uncertainty [49, 11, 29]. However, those methods need to compute gradient for each class, and thus they can hardly be applied on data sets with a large class number. Besides, recent works [50, 51] leverage NTK to analyze contextual bandit with streaming data, which are hard to be applied into our pool-based setting.

# 7 Conclusion

In this work, we bridge the gap between the theoretic findings of deep neural networks and real-world deep active learning applications. By exploring the connection between the generalization performance and the training dynamics, we propose a theory-driven method, *dynamicAL*, which selects samples to maximize training dynamics. We prove that the convergence speed of training and the generalization performance is (positively) strongly correlated under the ultra-wide condition and we show that maximizing the training dynamics will lead to a lower generalization error. Empirically, our work shows that *dynamicAL* not only consistently outperforms strong baselines across various setting, but also scales well on large deep learning models.

# 8 Acknowledgment

This work is supported by National Science Foundation (IIS-1947203, IIS-2117902, IIS-2137468, IIS-2134079, and CNS-2125626), a joint ACES-ICGA funding initiative via USDA Hatch ILLU-802-946, and Agriculture and Food Research Initiative (AFRI) grant no. 2020-67021-32799/project accession no.1024178 from the USDA National Institute of Food and Agriculture. The views and conclusions are those of the authors and should not be interpreted as representing the official policies of the funding agencies or the government.

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
