## A  APPENDIX: Derivation of Objectives

For the notational convenience, we use $f(x)$ to represent $f(x; \theta)$ in the Appendix.

### A.1  Training Dynamics for Cross-Entropy Loss

The partial derivative for softmax function can be defined with the following,

$$\frac{\partial \sigma^i(f(x))}{\partial f^j(x)} = \begin{cases} \sigma^i(f(x))\big(1 - \sigma^i(f(x))\big), & i = j, \\ -\sigma^i(f(x))\sigma^j(f(x)), & i \neq j \end{cases} \tag{24}$$

Then, we have:

$$\begin{aligned}
\frac{\partial \ell(f(x), y)}{\partial t} &= -\sum_i y^i \frac{\partial \log \sigma^i f(x)}{\partial \sigma^i(f(x))} \frac{\partial \sigma^i(f(x))}{\partial t} \\
&= -\sum_i y^i \frac{1}{\sigma^i(f(x))} \sum_j \frac{\partial \sigma^i(f(x))}{\partial f^j(x)} \frac{\partial f^j(x)}{\partial t} \\
&= -\sum_i y^i \sum_j \big(\mathbb{1}[i == j] - \sigma^j(f(x))\big) \frac{\partial f^j(x)}{\partial t} \\
&= -\sum_i \big(y^i - \sigma^i(f(x))\big) \nabla_\theta f^i(x) \nabla_t \theta
\end{aligned} \tag{25}$$

### A.2  Derivation for Cross-Entropy Loss

$$\begin{aligned}
\frac{\partial \ell(f(x), y)}{\partial \theta} &= \frac{\partial \ell}{\partial f(x)} \frac{\partial f(x)}{\partial \theta} = -\sum_i y^i \frac{1}{\sigma^i(f(x))} \frac{\partial \sigma^i(f(x))}{f(x)} \frac{\partial f(x)}{\partial \theta} \\
&= -\sum_i y^i \frac{1}{\sigma^i(f(x))} \sigma^i(f(x)) \sum_j \big(\mathbb{1}[i == j] - \sigma^j(f(x))\big) \frac{\partial f^j(x)}{\partial \theta} \\
&= \sum_j \big(\sigma^j(f(x)) - y^j\big) \frac{\partial f^j(x)}{\partial \theta}
\end{aligned} \tag{26}$$

### A.3  APPENDIX: Training Dynamics for Mean Squared Error

For the labeled data set $S$, we define the Mean Squared Error(MSE) as:

$$L_{MSE}(S) = \sum_{(x,y) \in S} \ell_{MSE}(f(x), y) = -\sum_{(x,y) \in S} \sum_{i \in [K]} \frac{1}{2}(f^i(x) - y^i)^2$$

Then the training loss dynamics for each sample can be defined as:

$$\frac{\partial \ell_{MSE}(f(x), y)}{\partial t} = -\sum_i \big(y^i - f^i(x)\big) \nabla_\theta f^i(x) \nabla_t \theta$$

Because neural networks are optimized by gradient descent, thus:

$$\nabla_t \theta = \theta_{t+1} - \theta_t = \sum_{(x,y) \in S} \frac{\partial \ell(f(x), y)}{\partial \theta} = \sum_{(x,y) \in S} \sum_j \big(f^j(x) - y^j\big) \frac{\partial f^j(x)}{\partial \theta}$$

Therefore, the training dynamics of MSE loss can be expressed as:

$$G_{MSE}(S) = -\frac{1}{\eta} \frac{\partial \sum_{(x,y) \in S} \ell_{MSE}(f(x), y)}{\partial t} = (f(X) - Y)^\top \mathcal{K}(X, X)(f(X) - Y)$$

### A.4   APPENDIX: Decomposition of the Change of Training Dynamics

According to the definition of training dynamics ( Equation (8) ), we have,

$$G(S) = \sum_{i,j} \sum_{(x_l,y_l)\in S} \left(\sigma^i(f(x_l;\theta)) - y_l^i\right) \sum_{(x_{l'},y_{l'})\in S} \nabla_\theta f^i(x_l;\theta)^\top \nabla_\theta f^j(x_{l'};\theta)\left(\sigma^j(f(x_{l'};\theta)) - y_{l'}^j\right)$$

$$G(S\cup\widehat{Q}) = \sum_{i,j} \sum_{(x,y)\in S\cup\widehat{Q}} \left(\sigma^i(f(x;\theta)) - y^i\right) \sum_{(x',y')\in S\cup\widehat{Q}} \nabla_\theta f^i(x;\theta)^\top \nabla_\theta f^j(x';\theta)\left(\sigma^j(f(x';\theta)) - y'^j\right)$$

The change of training dynamics, $\Delta(\widehat{Q}|S) = G(S \cup \widehat{Q}) - G(S)$, can be further simplified as:

$$
\begin{aligned}
\Delta(\widehat{Q}|S) &= G(S\cup\widehat{Q}) - G(S) \\
&= 2\sum_{i,j} \sum_{(x_u,\widehat{y}_u)\in\widehat{Q}} \left(\sigma^i(f(x_u;\theta)) - \widehat{y}_u^i\right) \sum_{(x_l,y_l)\in S} \nabla_\theta f^i(x_u;\theta)^\top \nabla_\theta f^j(x_l;\theta)\left(\sigma^j(f(x_l;\theta)) - y_l^j\right) \\
&+ \sum_{i,j} \sum_{(x_u,\widehat{y}_u)\in\widehat{Q}} \left(\sigma^i(f(x_u;\theta)) - \widehat{y}_u^i\right)\nabla_\theta f^i(x_u;\theta)^\top \nabla_\theta f^j(x_u;\theta)\left(\sigma^j(f(x_u;\theta)) - \widehat{y}_u^j\right) \\
&+ \sum_{i,j} \sum_{(x_u,\widehat{y}_u)\in\widehat{Q}} \left(\sigma^i(f(x_u;\theta)) - \widehat{y}_u^i\right) \sum_{(x_{u'},\widehat{y}_{u'})\in\widehat{Q},u'\neq u} \nabla_\theta f^i(x_{u'};\theta)^\top \nabla_\theta f^j(x_{u'};\theta)\left(\sigma^j(f(x_{u'};\theta)) - \widehat{y}_{u'}^j\right) \\
&= \sum_{(x_u,\widehat{y}_u)\in\widehat{Q}} \Delta(\{(x_u,\widehat{y}_u)\}|S) + \sum_{(x_u,\widehat{y}_u),(x_{u'},\widehat{y}_{u'})\in\widehat{Q}} d^i(x_u,\widehat{y}_u)^\top \mathcal{K}^{ij}(x_u,x_{u'}) d^i(x_{u'},\widehat{y}_{u'})
\end{aligned}
$$

### A.5   APPENDIX: Simplification of the Change of Training Dynamics

$$
\begin{aligned}
\Delta(\{(x_u,\widehat{y}_u)\}|S) =& 2\sum_{i,j} \sum_{(x_u,\widehat{y}_u)\in\widehat{Q}} \left(\sigma^i(f(x_u;\theta)) - \widehat{y}_u^i\right) \sum_{(x_l,y_l)\in S} \nabla_\theta f^i(x_u;\theta)^\top \nabla_\theta f^j(x_l;\theta)\left(\sigma^j(f(x_l;\theta)) - y_l^j\right) \\
&+ \sum_{i,j} \sum_{(x_u,\widehat{y}_u)\in\widehat{Q}} \left(\sigma^i(f(x_u;\theta)) - \widehat{y}_u^i\right)\nabla_\theta f^i(x_u;\theta)^\top \nabla_\theta f^j(x_u;\theta)\left(\sigma^j(f(x_u;\theta)) - \widehat{y}_u^j\right)
\end{aligned}
$$

The derivative of loss with respect to model parameters can be written as:

$$\frac{\partial \sum_{(x,y)\in S} \ell(f(x;\theta),y)}{\partial\theta} = \sum_{(x,y)\in S} \sum_{j\in[K]} \left(\sigma^j(f(x;\theta)) - y^j\right)\nabla_\theta f^j(x;\theta)$$

Therefore, the change of training dynamics caused by $\{(x_u,\widehat{y}_u)\}$ can be written as:

$$\Delta(\{(x_u,\widehat{y}_u)\}|S) = \|\nabla_\theta \ell(f(x_u;\theta),\hat{y}_u)\|^2 + 2\sum_{(x,y)\in S} \nabla_\theta \ell(f(x_u;\theta),\hat{y}_u)^\top \nabla_\theta \ell(f(x;\theta),y)$$

## B   APPENDIX: Proofs for Theoretical Analysis

### B.1   Proofs for Theorem 1

**Lemma 1** (Convergence Analysis with NTK, Theorem 4.1 of [13]). *Suppose $\lambda_0 = \lambda_{\min}(\boldsymbol{\Theta}) > 0$ for all subsets of data samples. For $\delta \in (0,1)$, if $m = \Omega(\frac{n^7}{\lambda_0^4\delta^4\epsilon^2})$ and $\eta = O(\frac{\lambda_0}{n^2})$, with probability at least $1 - \delta$, the network can achieve near-zero training error,*

$$\|Y - f_t(X;\theta(t))\|_2 = \sqrt{\sum_{k=1}^{K}\sum_{i=1}^{n}(1 - \eta\lambda_i)^{2t}(\vec{v}_i^\top Y^k)^2} \pm \epsilon \tag{27}$$

*where $n$ denotes the number of training samples and $m$ denotes the width of hidden layers. The NTK $\boldsymbol{\Theta} = V^\top \Lambda V$ with $\Lambda = \{\lambda_i\}_{i=1}^{n}$ is a diagonal matrix of eigenvalues and $V = \{\vec{v}_i\}_{i=1}^{n}$ is a unitary matrix.*

*Proof.* According to [13], if $m = \Omega(\frac{n^7}{\lambda_0^4 \delta^4 \epsilon^2})$ and learning ratio $\eta = O(\frac{\lambda_0}{n^2})$, then with probability at least $1 - \delta$ over the random initialization, we have, $\|Y_l - f_t(X; \theta(t))\|_2 = \sqrt{\sum_{k=1}^K \sum_{i=1}^n (1 - \eta\lambda_i)^{2t} (v_i^\top Y_l^k)^2} \pm \epsilon$. We decompose the NTK using $\mathbf{\Theta} = V^\top \Lambda V$ with $\Lambda = \{\lambda_i\}_{i=1}^n$ a diagonal matrix of eigenvalues and $V = \{v_i\}_{i=1}^n$ a unitary matrix. At each training step in active learning, the labeled samples will be updated by $S = S \cup \overline{Q}$. We can apply the convergence result in each of this step and achieve near zero error. $\qquad \square$

**Theorem 1** (Relationship between convergence rate and alignment). *Under the same assumptions as in Lemma 1, the convergence rate described by $\mathcal{E}_t$ satisfies,*

$$\mathrm{Tr}[Y^\top Y] - 2t\eta\mathcal{A}(X, Y) \leq \mathcal{E}_t^2(X, Y) \leq \mathrm{Tr}[Y^\top Y] - \eta\mathcal{A}(X, Y) \tag{28}$$

*Proof.* We first prove the inequality on the right hand side. It is easy to see that $(1-\eta\lambda_i)^{2t} \leq (1-\eta\lambda_i)$ for each $\lambda_i$ and $t \geq 1$, based on the fact that $\forall \lambda_i, 0 \leq 1 - \eta\lambda_i \leq 1$. Then we can obtain,

$$\mathcal{E}_t(X, Y) = \sqrt{\sum_{k=1}^K \sum_{i=1}^n (1 - \eta\lambda_i)^{2t} (v_i^\top Y^k)^2} \leq \sqrt{\sum_{k=1}^K \sum_{i=1}^n (1 - \eta\lambda_i)(v_i^\top Y^k)^2}$$

$$= \sqrt{\mathrm{Tr}[Y^\top (I - \eta\mathbf{\Theta})Y]} = \sqrt{\mathrm{Tr}[Y^\top Y] - \eta\mathcal{A}(X, Y)}$$

Then we use Bernoulli's inequality to prove the inequality on the left hand side. Bernoulli's inequality states that, $(1 + x)^r \geq 1 + rx$, for every integer $r \geq 0$ and every real number $x \geq -1$. It is easy to check that $(-\eta\lambda_i) \geq -1, \forall \lambda_i$. Therefore,

$$\mathcal{E}_t(X, Y) = \sqrt{\sum_{k=1}^K \sum_{i=1}^n (1 - \eta\lambda_i)^{2t} (v_i^\top Y^k)^2} \geq \sqrt{\sum_{k=1}^K \sum_{i=1}^n (1 - 2t\eta\lambda_i)(v_i^\top Y^k)^2}$$

$$= \sqrt{\mathrm{Tr}[Y^\top (I - 2t\eta\mathbf{\Theta})Y]} = \sqrt{\mathrm{Tr}[Y^\top Y] - 2t\eta\mathcal{A}(X, Y)}$$

$$\square$$

### B.2 Proof for Theorem 2

**Lemma 2** (Generalization bound with NTK, Theorem 5.1 of [13]). *Suppose data $S = \{(x_i, y_i)\}_{i=1}^n$ are i.i.d. samples from a non-degenerate distribution $p(x, y)$, and $m \geq \mathrm{poly}(n, \lambda_0^{-1}, \delta^{-1})$. Consider any loss function $\ell : \mathbb{R} \times \mathbb{R} \to [0, 1]$ that is 1-Lipschitz, with probability at least $1 - \delta$ over the random initialization, the network trained by gradient descent for $T \geq \Omega(\frac{1}{\eta\lambda_0} \log \frac{n}{\delta})$ iterations has population risk $\mathcal{L}_p = \mathbb{E}_{(x,y)\sim p(x,y)}[\ell(f_T(x), y)]$ that is bounded as follows:*

$$\mathcal{L}_p \leq \sqrt{\frac{2\,\mathrm{Tr}[Y^\top \mathbf{\Theta}^{-1}(X, X)Y]}{n}} + O\left(\sqrt{\frac{\log \frac{n}{\lambda_0\delta}}{n}}\right). \tag{29}$$

*Proof.* We first show that the generalization bound regrading our method on ultra-wide networks. The distance between weights of trained networks and their initialization values can be bounded as, $\|w_r(t) - w_r(0)\| = O(\frac{n}{\sqrt{m}\lambda_0\sqrt{\delta}})$. We then give a bound on the $\|W(t) - W(0)\|_F$, where $W = \{w_1, w_2, \dots\}$ is the set of all parameters. We definite $Z = \frac{\partial f(t)}{\partial W(t)}$, then the update function is given by $W(t + 1) = W(t) - \eta Z(Z^\top W(t) - Y)$. Summing over all the time step $t = 0, 1, \dots$, we can obtain that $W(\infty) - W(0) = \sum_{t=0}^\infty \eta Z(I - \eta\mathbf{\Theta})y = Z\mathbf{\Theta}^{-1}Y$. Thus the distance can be measured by $\|W(\infty) - W(0)\|_F^2 = \mathrm{Tr}[Y^\top \mathbf{\Theta}^{-1}Y]$.

Then the key step is to apply Rademacher complexity. Given $R > 0$, with probability at least $1 - \delta$, simultaneously for every $B > 0$, the function class $\mathcal{F}_{B,R} = \{f : \|w_r(t) - w_r(0)\| \leq R \ (\forall r \in m), \|W(\infty) - W(0)\|_F^2 \leq B\}$ has empirical Rademacher complexity bounded as,

$$\mathcal{R}_S(\mathcal{F}_{B,R}) = \frac{1}{n}\mathbb{E}_{\epsilon_i \in \{\pm 1\}^n}\left[\sup_{f \in \mathcal{F}_{B,R}} \sum_{i=1}^n \epsilon_i f(x_i)\right] \leq \frac{B}{\sqrt{2n}}\left(1 + (\frac{2\log\frac{2}{\delta}}{m})^{1/4}\right) + 2R^2\sqrt{m} + R\sqrt{2\log\frac{2}{\delta}}$$

where $B = \sqrt{\mathrm{Tr}[Y^\top \Theta^{-1}(X,X)Y]}$, and $R = \frac{n}{\sqrt{m}\lambda_0\sqrt{\delta}}$.

Finally, Rademacher complexity directly gives an upper bound on generalization error [52], $\sup_{f \in \mathcal{F}}\{\mathcal{L}_p(f) - \mathcal{L}_S(f)\} \leq 2\mathcal{R}_S + 3c\sqrt{\frac{\log(2/\delta)}{2n}}$, where $\mathcal{L}_S(f) \leq \frac{1}{\sqrt{n}}$. Based on this, we apply a union bound over a finite set of different $i$'s. Then with probability at least $1 - \delta/3$ over the sample $S$, we have $\sup_{f \in \mathcal{F}_{R,B_i}}\{\mathcal{L}_p(f) - \mathcal{L}_S(f)\} \leq 2\mathcal{R}_S(\mathcal{F}_{B_i,R}) + O(\sqrt{\frac{\log \frac{n}{\lambda_0\delta}}{n}})$, $\forall i \in \{1, 2, \ldots, O(\frac{n}{\lambda_0})\}$. Taking a union bound, we know that with probability at least $1 - \frac{2}{3}\delta$ over the sample $S$, we have, $f_T \in \mathcal{F}_{B_i^*,R}$ for some $i^*$, $\mathcal{R}_S(\mathcal{F}_{B_i^*,R}) \leq \sqrt{\frac{\mathrm{Tr}[Y^\top\Theta^{-1}(X,X)Y]}{2n}} + \frac{2}{\sqrt{n}}$ and $\sup_{f_T \in \mathcal{F}_{B_i^*,R}}\{\mathcal{L}_p(f_T) - \mathcal{L}_S(f_T)\} \leq 2\mathcal{R}_S(\mathcal{F}_{B_i^*,R}) + O(\sqrt{\frac{\log \frac{n}{\lambda_0\delta}}{n}})$. These together can imply,

$$\mathcal{L}_p(f) \leq \frac{1}{\sqrt{n}} + 2\mathcal{R}_S(\mathcal{F}_{B_i^*,R}) + O(\sqrt{\frac{\log \frac{n}{\lambda_0\delta}}{n}}) \leq \sqrt{\frac{2\,\mathrm{Tr}[Y^\top\Theta^{-1}(X,X)Y]}{n}} + O\left(\sqrt{\frac{\log \frac{n}{\lambda_0\delta}}{n}}\right).$$

More proof details can be found in [13]. $\qquad\square$

**Theorem 2** (Relationship between the generalization bound and alignment). *Under the same assumptions as in Lemma* ([2](#))*, if we define the generalization upper bound as* $\mathcal{B}(X,Y) = \sqrt{\frac{2\,\mathrm{Tr}[Y^\top\Theta^{-1}Y]}{n}}$*, then it can be bounded with the alignment as follows,*

$$\frac{\mathrm{Tr}^2[Y^\top Y]}{\mathcal{A}(X,Y)} \leq \frac{n}{2}\mathcal{B}^2(X,Y) \leq \frac{\lambda_{max}}{\lambda_{min}}\frac{\mathrm{Tr}^2[Y^\top Y]}{\mathcal{A}(X,Y)} \tag{30}$$

*Proof.* We first expand the following expression:

$$\frac{n}{2}\mathcal{B}^2(X,Y)\mathcal{A}(X,Y) = \sum_{k=1}^{K}\sum_{i=1}^{n}\lambda_i(v_i^\top Y^k)^2 \sum_{k=1}^{K}\sum_{i=1}^{n}\frac{1}{\lambda_i}(v_i^\top Y^k)^2$$

Then we use this expansion to prove the inequality on the left hand side,

$$\sum_{k=1}^{K}\sum_{i=1}^{n}\lambda_i(v_i^\top Y^k)^2 \sum_{k=1}^{K}\sum_{i=1}^{n}\frac{1}{\lambda_i}(v_i^\top Y^k)^2 = \sum_{k=1}^{K}\sum_{k'=1}^{K}\left(\sum_{i=1}^{n}\lambda_i(v_i^\top Y^k)^2 \sum_{i=1}^{n}\frac{1}{\lambda_i}(v_i^\top Y^{k'})^2\right)$$

$$\geq \sum_{k=1}^{K}\sum_{k'=1}^{K}\left(\sum_{i=1}^{n}(v_i^\top Y^k)^2 \sum_{i=1}^{n}(v_i^\top Y^{k'})^2\right) = \left(\sum_{k=1}^{K}Y^{k^\top}V^\top V Y^k\right)\left(\sum_{k=1}^{K}Y^{k^\top}V^\top V Y^k\right)$$

$$= \mathrm{Tr}^2[Y^\top Y]$$

The second line is due to quadratic mean is greater or equal to geometric mean. Finally, we prove the inequality on the right hand side,

$$\sum_{k=1}^{K}\sum_{i=1}^{n}\lambda_i(v_i^\top Y^k)^2 \sum_{k=1}^{K}\sum_{i=1}^{n}\frac{1}{\lambda_i}(v_i^\top Y^k)^2 = \sum_{k=1}^{K}\sum_{k'=1}^{K}\left(\sum_{i=1}^{n}\lambda_i(v_i^\top Y^k)^2 \sum_{i=1}^{n}\frac{1}{\lambda_i}(v_i^\top Y^{k'})^2\right)$$

$$\leq \sum_{k=1}^{K}\sum_{k'=1}^{K}\frac{\lambda_{max}}{\lambda_{min}}\left(\sum_{i=1}^{n}(v_i^\top Y^k)^2 \sum_{i=1}^{n}(v_i^\top Y^{k'})^2\right) = \frac{\lambda_{max}}{\lambda_{min}}\left(\sum_{k=1}^{K}Y^{k^\top}V^\top V Y^k\right)\left(\sum_{k=1}^{K}Y^{k^\top}V^\top V Y^k\right)$$

$$= \frac{\lambda_{max}}{\lambda_{min}}\mathrm{Tr}^2[Y^\top Y]$$

$\qquad\square$

## B.3 Derivation for Maximum Mean Discrepancy

The difference between truth risk over $p(x)$ and $q(x)$ can be defined as,

$$\mathcal{L}_p - \mathcal{L}_q = \int_x g(x)p(x)dx - \int_x g(x)q(x)dx$$

where $g(x) = \int_y \ell(f(x;\theta), y)p(y|x)dy$. Follow [34], we assume that the prediction functions have bounded norm $\|f\|_F$. Thus, the function $g$ is bounded. By given the loss function, $g$ is also measurable. Then, $\exists \hat{g} \in \mathcal{C}(x)$, such that,

$$\int_x g(x)p(x)dx - \int_x g(x)q(x)dx = \int_x \hat{g}(x)p(x)dx - \int_x \hat{g}(x)q(x)dx$$

$$\leq \sup_{\hat{g}\in\mathcal{C}(x)} \int_x \hat{g}(x)p(x)dx - \int_x \hat{g}(x)q(x)dx = \text{MMD}\big(p(x), q(x), \mathcal{C}\big)$$

where $\mathcal{C}(x)$ is the function class of bounded and continuous functions of $x$. To make the MMD term be measurable, we empirically restrict the MMD on a reproducing kernel Hilbert space (RKHS) with the characteristic kernel $\mathcal{H}_\Theta$. Following [53], we know that the relationship between the true MMD and the empirical MMD is,

$$P\Big(\big|\text{MMD}\big(p(x), q(x), \mathcal{C}\big) - \text{MMD}(S_0, S, \mathcal{H}_\Theta)\big| \geq \epsilon + 2(\sqrt{\frac{C}{n_0}} + \sqrt{\frac{C}{n}})\Big) \quad \leq 2e^{\frac{-\epsilon^2 n_0 n}{2C(n_0+n)}}$$

where $\text{MMD}(S_0, S, \mathcal{H}_\Theta)$ is the empirical measure for $\text{MMD}(p(x), q(x), \mathcal{H}_\Theta)$. Slightly overloading the notation, we denote $S \sim q(x)$, which may not be i.i.d., and the initial label set $S_0 \sim p(x)$. Then, in the active learning setting, $S_0 \subseteq S$. Further, we denote $|S_0| = n_0, |S| = n$ and $\forall x, x' \in S, \Theta(x, x') \leq C$. Therefore, we have, $\sqrt{\frac{C}{n}} + \sqrt{\frac{C}{n_0}} \geq 2\sqrt{\frac{C}{n}}$. For constant factor $\gamma = \frac{M}{M+B}$, we have the following inequality,

$$P\big(\text{MMD}\big(p(x), q(x), \mathcal{C}\big) \geq \text{MMD}(S_0, S, \mathcal{H}_\Theta) + \epsilon + 4\sqrt{\frac{C}{n}}\big) \leq 2e^{\frac{-\gamma\epsilon^2 n}{4C}}$$

Denoting $2e^{\frac{-\gamma\epsilon^2 n}{4C}} = \delta/2$, then we have $\epsilon = \sqrt{\frac{4C\ln(4/\delta)}{\gamma n}}$. Combining all the above results, we show that with probability at least $1 - \delta$, the following inequality holds:

$$\mathcal{L}_p - \mathcal{L}_q \leq \text{MMD}(S_0, S, \mathcal{H}_\Theta) + 4\sqrt{\frac{C}{n}} + \sqrt{\frac{4C\ln(4/\delta)}{\gamma n}}$$

Then, we can get,

$$\mathcal{L}_p - \mathcal{L}_q \leq \text{MMD}(S_0, S, \mathcal{H}_\Theta) + O\left(\sqrt{\frac{C\ln(1/\delta)}{n}}\right)$$

## C  APPENDIX: More details of experimental settings

### C.1  Implementation Detail

For simple CNN model, we utilize the same architecture used in Pytorch CIFAR10 Image Classification Tutorial [1]. For ResNet model, we use the Pytorch Offical implementation of ResNet-18 [2] and set the output dimension to the number of classes. For VGG model, we use the Pytorch Offical implementation of VGG-11 [3]. Besides, we leverage the library BackPACK [54] to collect the gradient of samples in batch.

We keep a constant learning rate of 0.001 for all three datasets and all three models. All the codes mentioned above use the MIT license. All experiments are done with four Tesla V100 SXM2 GPUs and a 12-core 2.2GHz CPU.

---

[1] https://pytorch.org/tutorials/beginner/blitz/cifar10_tutorial.html
[2] https://github.com/pytorch/vision/blob/main/torchvision/models/resnet.py
[3] https://github.com/pytorch/vision/blob/main/torchvision/models/vgg.py

## C.2 Computation of Acquisition Function

The acquisition function employed by *dynamicAL* can be written as the Equation 15. Furthermore, we simplify it into the following form:

$$\Delta(\{(x_u, \widehat{y_u})\}|S) = \|\nabla_\theta \ell(f(x_u; \theta), \hat{y}_u)\|^2 + 2\nabla_\theta \ell(f(x_u; \theta), \hat{y}_u)^\top \nabla_\theta \ell(f(X_S; \theta), Y_S). \tag{31}$$

where $\nabla_\theta \ell(f(X_S; \theta), Y_S) = \sum_{(x,y) \in S} \nabla_\theta \ell(f(x; \theta), y)$. The computational requirement of the Equation 31 is mainly composed of two parts, the computation of gradient and the computation of the inner product. While PyTorch [55] can compute efficiently batch gradients, BackPACK [54] optimizes the computation of individual gradient and compute the gradient norm, sample per sample, at almost no time overhead. Thus, the acquisition function can be computed at low computational costs. Note, the efficiency of BackPACK has been verified by several recent works with extensive experiments[56, 57].

# D   APPENDIX: Verification Experiments under Ultra-wide Condition

## D.1   Experiment Setting and Computational Detail for the Empirical Comparison between NTK and MMD

**Experiment Setting** For the verification experiment shown in Figure 1, we employ a simple CNN as the target model, in which there are three convolutional layers following with global average pooling layer, on the CFAIR10 data set. Note, this CNN architecture is widely used in NTK analysis works [33, 58]. To make the verification experiment close to the application setting, we keep size of initial labeled set and query batch size same as what we used in Section 5.

**Computational Detail** We follow [35] to compute the MMD with NTK kernel. The MMD term, $\mathrm{MMD}(p(x), q(x), \mathcal{H}_\Theta)$, can be simplified into the following form:

$$\mathrm{MMD}^2(p(x), q(x)) = \mathbb{E}[\Theta(x_i, x_j) + \Theta(x'_i, x'_j) - 2\Theta(x_i, x'_j)] \tag{32}$$

where $x_i, x_j \sim p(x)$ and $x'_i, x'_j \sim q(x)$. Then, we define set $S_0$ as $\{x_1, ..., x_{n_0}\} \sim p(x)$ and set $S$ as $\{x'_1, ..., x'_n\} \sim q(x)$, where $n_0 \leq n$. The $\mathrm{MMD}^2(S_0, S)$ is an unbiased estimation for $\mathrm{MMD}^2(p(x), q(x))$, can we explicitly computed by:

$$\begin{aligned}
\mathrm{MMD}^2(S_0, S) &= \frac{1}{m^2 - m}a + \frac{1}{n^2 - n}b - \frac{2}{m(n-1)}c \\
a &= \left( \sum_{i,j}^m \Theta(x_i, x_j) - \sum_i^m \Theta(x_i, x_i) \right) \\
b &= \left( \sum_{i,j}^n \Theta(x'_i, x'_j) - \sum_i^n \Theta(x'_i, x'_i) \right) \\
c &= \left( \sum_i^m \sum_j^n \Theta(x_i, x'_j) - \sum_{i,j}^m \Theta(x_i, x'_i) \right)
\end{aligned} \tag{33}$$

Therefore, the MMD term of Equation (23), $\mathrm{MMD}(S_0, S, \mathcal{H}_\Theta)$, can be empirically approximated by $\sqrt{\mathrm{MMD}^2(S_0, S)}$.

## D.2   Experiment for the Correlation Study between Training Dynamics and Alignment

**Experiment Setting.** For the verification experiment shown in Figure 2, we also use the simple CNN on CIFAR10. And to keep consistent with the application setting, we set $|S| = 500$ and $|\overline{Q}| = 250$. The $\overline{Q}$ is randomly sampled from the unlabeled set and the labeled set $S$ is fixed. We independently sample $\overline{Q}$ 150 times to compute the correlation between between $G_{MSE}(S \cup \overline{Q})$ and $\mathcal{A}(X\|X_Q, Y\|Y_Q)$.

**Correlation between Training Dynamics computed with pseudo-labels and Alignment.**

In Figure 2, we compute the training dynamics with the ground-truth label. To study the effect of pseudo-labels, we further provide the relation between training dynamics computed with pseudo-labels $G_{MSE}(S \cup \overline{Q})$ and alignment $\mathcal{A}(X\|X_Q, Y\|Y_Q)$, in which we compute the pseudo-labels with $\Theta(X_Q, X)^\top \Theta(X, X)^{-1}Y$. The result is shown in the Figure 6. Note that we keep hyperparameters the same as previously described. Compared with Figure 2, we find that the positive relationship between $\mathcal{A}$ and the $G$ computed with ground-truth labels is stronger than the $G$ computed with pseudo-labels. The result is aligned with our expectations, because the extra noise is introduced by the pseudo-labels. But, the Kendall $\tau$ coefficient still achieves 0.46 for $\mathcal{A}$ and the $G$ computed with pseudo-labels which indicates the utility of using $G$ calculating with pseudo-labels as the acquisition function to query samples.

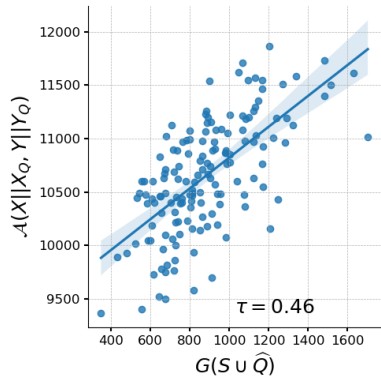

Figure 6: Relation between Alignment and Training Dynamics computed with the pseudo-label.

### D.3 Correlation Study
### between Training Dynamics and Generalization Bound

We present the relation between the training dynamics and the generalization bound in Figure 7. Same as the previous, we set $|S| = 500$ and $|\overline{Q}| = 250$ and the $\overline{Q}$ is randomly sampled from the unlabeled set. The result shows that with the increase of $G$, $B$ will decrease. This empirical observation is aligned with our expectation, because Theorem 2 indicates that the alignment $A$ is inverse proportional to $B$ and Figure 2 tells us that the $G$ is proportional to $A$. Besides, the $\tau$ achieves -0.253 which indicates that the $A$ is moderately inverse proportional to $B$ [59].

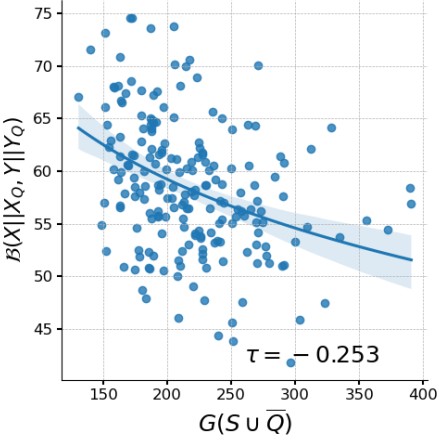

Figure 7: Relationship between Training Dynamics and Generalization.

## E  APPENDIX: More details of experimental results

### E.1  Baselines

1. Random: Unlabeled data are randomly selected at each round.

2. Coreset: This method performs a clustering over the last hidden representations in the network, and calculates the minimum distance between each candidate sample's embedding and embeddings of labeled samples. Then data samples with the maximum distances are selected. [60].

3. Confidence Sampling (Conf): The method selects $b$ examples with smallest predicted class probability $\max_i^K f^i(x; \theta)$ [61].

4. Margin Sampling (Marg): The bottom $b$ examples sorted according to the example's multi-class margin are selected. The margin is defined as $f^i(x;\theta) - f^j(x;\theta)$, where $i$ and $j$ are the indices of the largest and second largest entries of $f(x;\theta)$ [62].

5. Entropy: Top $b$ samples are selected according to the entropy of the example's predictive class probability distribution, the entropy is defined as $H((f^i(x;\theta))_{i=1}^K)$, where $H(p) = \sum_i^K p_i \ln \frac{1}{p_i}$ [61].

6. Active Learning by Learning (ALBL): The bandit-style meta-active learning algorithm combines Coreset and Conf [63].

7. Batch Active learning by Diverse Gradient Embeddings (BADGE): $b$ samples are selected by using k-means++ seeding on the gradients of the final layer, in order to query by uncertainty and diversity. [11].

## E.2 Experiment Results

The results for ResNet18, VGG11, and vanilla CNN on CIFAR10, SVHN, and Caltech101 with different batch sizes have been shown in the Figure 3 and 8. Note, for the large batch size setting ($b = 1000$) on Caltech101, we set the number of query round $R = 4$, in which 49.2% images will be labeled after 4 rounds.

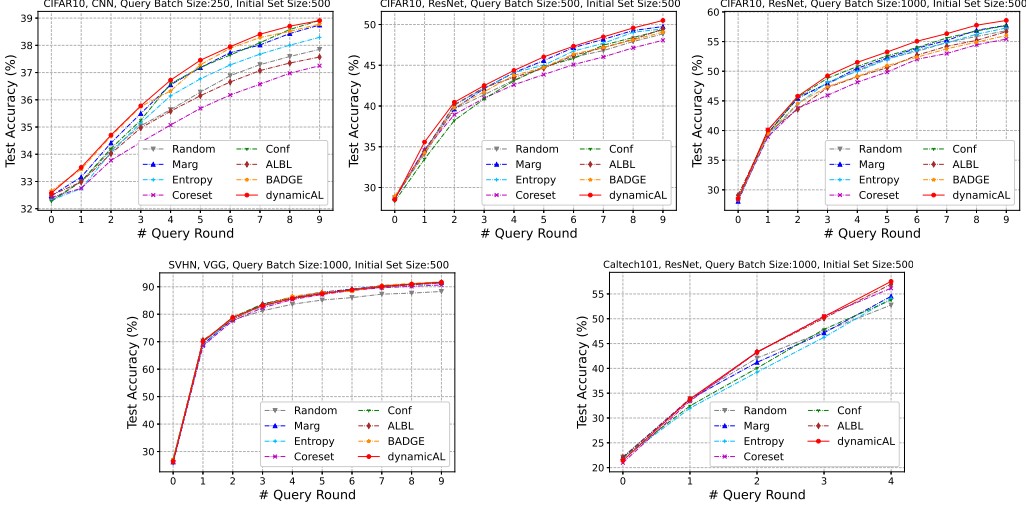

Figure 8: The evaluation results for different active learning methods under a range of conditions.

## E.3 Numerical Result of Main Experiments

For the the main experiments, we report the means and standard deviations of active learning performance under different settings in the the following tables.

Table 2: CIFAR10, CNN, Query Batch Size:250, Initial Set Size:500

| | RANDOM | MARG | ENTROPY | CORESET | CONF | ALBL | BADGE | *dynamicAL* |
|---|---|---|---|---|---|---|---|---|
| 0 | 32.32%±1.308% | 32.48%±1.286% | 32.32%±1.269% | 32.41%±1.281% | 32.27%±1.266% | 32.50%±1.263% | 32.66%±1.182% | 32.57%±1.423% |
| 1 | 33.00%±1.175% | 33.16%±1.165% | 32.74%±1.617% | 32.75%±1.306% | 33.00%±1.703% | 32.98%±1.184% | 33.45%±1.813% | 33.52%±1.311% |
| 2 | 34.14%±1.322% | 34.41%±1.130% | 34.06%±1.546% | 33.77%±1.011% | 34.21%±1.426% | 34.02%±1.392% | 34.66%±1.483% | 34.70%±1.019% |
| 3 | 35.05%±1.508% | 35.50%±1.301% | 35.16%±1.679% | 34.44%±0.937% | 35.25%±1.344% | 34.97%±1.227% | 35.75%±1.024% | 35.78%±1.115% |
| 4 | 35.64%±1.945% | 36.55%±1.249% | 36.14%±1.646% | 35.08%±1.396% | 36.59%±1.508% | 35.58%±1.177% | 36.33%±0.791% | 36.72%±0.716% |
| 5 | 36.28%±1.124% | 37.18%±1.547% | 36.77%±1.004% | 35.68%±1.390% | 37.19%±1.063% | 36.15%±1.311% | 37.29%±1.126% | 37.45%±1.573% |
| 6 | 36.88%±1.568% | 37.73%±1.546% | 37.28%±1.983% | 36.18%±1.419% | 37.65%±2.062% | 36.65%±1.111% | 37.90%±1.988% | 37.95%±1.414% |
| 7 | 37.29%±1.605% | 38.01%±0.874% | 37.67%±1.723% | 36.57%±1.346% | 38.09%±1.174% | 37.07%±1.731% | 38.28%±1.474% | 38.41%±1.295% |
| 8 | 37.59%±1.848% | 38.43%±1.675% | 38.01%±1.601% | 36.98%±0.748% | 38.58%±1.556% | 37.35%±1.135% | 38.51%±1.091% | 38.70%±1.291% |
| 9 | 37.85%±1.789% | 38.75%±1.550% | 38.29%±1.312% | 37.25%±1.527% | 38.91%±1.902% | 37.57%±1.170% | 38.78%±0.776% | 38.91%±1.358% |

## Table 3: CIFAR10, CNN, Query Batch Size:500, Initial Set Size:500

|   | RANDOM | MARG | ENTROPY | CORESET | CONF | ALBL | BADGE | *dynamicAL* |
|---|--------|------|---------|---------|------|------|-------|-------------|
| 0 | 32.26%±1.164% | 32.31%±1.441% | 32.29%±1.397% | 32.54%±1.331% | 32.32%±1.288% | 32.41%±1.432% | 32.49%±1.320% | 32.37%±1.049% |
| 1 | 34.87%±1.286% | 34.89%±1.575% | 34.58%±1.664% | 33.84%±1.368% | 34.75%±1.503% | 34.08%±1.368% | 34.44%±1.230% | 34.88%±1.557% |
| 2 | 36.45%±0.842% | 36.69%±1.456% | 36.50%±1.463% | 35.96%±1.667% | 36.73%±1.744% | 35.62%±1.536% | 36.41%±1.175% | 36.78%±1.253% |
| 3 | 37.16%±0.767% | 37.99%±1.356% | 37.30%±1.221% | 36.20%±1.086% | 38.12%±1.663% | 36.55%±1.327% | 37.56%±1.284% | 38.30%±1.152% |
| 4 | 37.89%±0.880% | 39.15%±1.056% | 38.23%±0.878% | 36.73%±1.011% | 39.10%±1.336% | 37.20%±1.381% | 38.49%±1.238% | 39.37%±0.708% |
| 5 | 38.59%±0.861% | 39.98%±1.562% | 39.01%±1.278% | 37.33%±1.373% | 39.81%±1.402% | 37.80%±1.560% | 39.61%±1.219% | 40.09%±0.940% |
| 6 | 39.15%±1.108% | 40.70%±1.391% | 39.68%±1.315% | 37.97%±1.393% | 40.47%±1.126% | 38.47%±1.270% | 40.55%±1.066% | 40.75%±1.671% |
| 7 | 39.51%±1.219% | 40.99%±1.217% | 40.09%±1.408% | 38.53%±1.600% | 41.05%±1.448% | 39.11%±1.385% | 40.97%±0.814% | 41.21%±1.433% |
| 8 | 39.90%±0.807% | 41.39%±1.614% | 40.39%±1.357% | 39.06%±1.156% | 41.30%±1.865% | 39.55%±1.595% | 41.27%±1.409% | 41.59%±1.013% |
| 9 | 40.17%±1.170% | 41.64%±1.287% | 40.71%±0.739% | 39.43%±0.892% | 41.55%±1.341% | 39.95%±1.299% | 41.41%±0.949% | 41.78%±0.645% |

## Table 4: CIFAR10, ResNet, Query Batch Size:500, Initial Set Size:500

|   | RANDOM | MARG | ENTROPY | CORESET | CONF | ALBL | BADGE | *dynamicAL* |
|---|--------|------|---------|---------|------|------|-------|-------------|
| 0 | 28.75%±1.780% | 28.75%±0.369% | 28.63%±1.394% | 28.63%±1.120% | 28.31%±1.011% | 28.75%±0.957% | 28.95%±1.040% | 28.52%±0.686% |
| 1 | 34.11%±3.088% | 34.62%±2.022% | 34.42%±0.849% | 34.57%±0.992% | 33.49%±1.269% | 34.42%±2.077% | 34.26%±1.740% | 35.58%±2.858% |
| 2 | 39.63%±2.157% | 39.63%±0.313% | 40.08%±1.022% | 38.94%±1.408% | 38.23%±1.454% | 40.16%±2.574% | 39.78%±1.384% | 40.46%±0.959% |
| 3 | 41.38%±2.357% | 42.15%±0.810% | 42.18%±1.271% | 40.96%±0.961% | 40.87%±0.860% | 42.26%±2.347% | 41.74%±1.230% | 42.51%±0.799% |
| 4 | 43.18%±1.809% | 44.09%±1.165% | 44.09%±1.150% | 42.60%±1.094% | 43.10%±1.325% | 43.52%±3.064% | 43.76%±1.364% | 44.36%±0.980% |
| 5 | 44.73%±2.253% | 45.57%±1.115% | 45.00%±0.731% | 43.86%±1.369% | 44.83%±1.388% | 44.64%±3.097% | 44.73%±1.675% | 46.02%±0.754% |
| 6 | 46.00%±2.193% | 47.17%±0.929% | 46.74%±1.118% | 45.08%±1.549% | 45.83%±1.426% | 46.22%±2.601% | 46.38%±1.607% | 47.34%±1.027% |
| 7 | 46.80%±2.134% | 48.18%±1.230% | 47.69%±1.253% | 46.02%±1.589% | 47.47%±1.424% | 47.18%±2.384% | 47.17%±1.404% | 48.48%±1.452% |
| 8 | 47.91%±1.722% | 49.26%±0.652% | 49.05%±1.113% | 47.14%±1.880% | 48.40%±1.178% | 48.18%±2.503% | 48.11%±1.049% | 49.58%±1.673% |
| 9 | 48.84%±1.584% | 49.75%±1.341% | 49.46%±1.282% | 48.07%±1.480% | 49.35%±1.269% | 49.45%±2.529% | 49.06%±0.850% | 50.50%±1.301% |

## Table 5: CIFAR10, ResNet, Query Batch Size:1000, Initial Set Size:500

|   | RANDOM | MARG | ENTROPY | CORESET | CONF | ALBL | BADGE | *dynamicAL* |
|---|--------|------|---------|---------|------|------|-------|-------------|
| 0 | 28.34%±1.465% | 28.07%±2.604% | 28.24%±1.756% | 28.41%±0.722% | 29.05%±1.137% | 29.06%±0.847% | 28.43%±1.176% | 28.48%±2.062% |
| 1 | 40.08%±0.329% | 39.57%±1.551% | 39.09%±2.180% | 38.95%±1.047% | 39.50%±2.340% | 39.67%±1.489% | 39.46%±3.020% | 40.09%±1.795% |
| 2 | 45.63%±1.253% | 45.43%±0.444% | 44.48%±1.823% | 43.78%±0.986% | 45.62%±1.882% | 43.58%±1.329% | 44.55%±3.654% | 45.77%±2.290% |
| 3 | 47.90%±1.257% | 47.96%±0.735% | 48.15%±2.509% | 45.93%±0.682% | 48.82%±1.797% | 47.24%±1.926% | 47.39%±4.189% | 49.22%±1.704% |
| 4 | 50.13%±1.207% | 50.49%±0.807% | 49.97%±2.819% | 48.14%±0.566% | 50.79%±1.870% | 49.05%±1.831% | 49.13%±4.053% | 51.50%±1.925% |
| 5 | 52.14%±1.517% | 52.24%±0.781% | 52.00%±2.762% | 49.85%±1.075% | 52.59%±2.202% | 50.59%±1.636% | 50.94%±3.628% | 53.24%±1.927% |
| 6 | 53.33%±1.300% | 53.87%±0.635% | 53.57%±3.123% | 52.01%±0.772% | 53.99%±2.390% | 52.69%±1.599% | 52.36%±3.924% | 55.06%±1.697% |
| 7 | 54.84%±1.238% | 55.19%±1.136% | 54.79%±3.144% | 52.99%±1.147% | 55.60%±2.002% | 54.20%±1.685% | 53.77%±3.985% | 56.33%±1.613% |
| 8 | 55.86%±1.161% | 56.90%±0.732% | 56.23%±3.182% | 54.45%±0.821% | 56.79%±2.033% | 55.20%±1.868% | 54.91%±4.104% | 57.76%±1.796% |
| 9 | 56.84%±0.979% | 57.73%±0.500% | 57.29%±3.225% | 55.42%±0.954% | 57.70%±2.042% | 56.67%±1.783% | 56.02%±3.935% | 58.56%±1.574% |

## Table 6: SVHN, VGG, Query Batch Size:500, Initial Set Size:500

|   | RANDOM | MARG | ENTROPY | CORESET | CONF | ALBL | BADGE | *dynamicAL* |
|---|--------|------|---------|---------|------|------|-------|-------------|
| 0 | 25.94%±7.158% | 26.15%±6.290% | 26.41%±8.994% | 25.83%±5.845% | 26.52%±7.489% | 25.31%±5.030% | 26.38%±9.100% | 26.30%±5.505% |
| 1 | 61.23%±4.812% | 63.93%±3.127% | 59.02%±4.724% | 57.02%±3.672% | 61.99%±2.613% | 62.14%±5.531% | 58.70%±5.615% | 63.01%±12.293% |
| 2 | 71.35%±2.364% | 74.08%±0.933% | 71.25%±1.459% | 67.95%±2.870% | 73.31%±2.828% | 71.75%±2.555% | 74.74%±2.978% | 74.10%±4.557% |
| 3 | 76.34%±1.626% | 79.17%±1.064% | 76.74%±1.521% | 73.76%±2.844% | 78.02%±1.939% | 77.65%±1.518% | 77.75%±2.100% | 79.20%±2.651% |
| 4 | 78.86%±1.378% | 82.18%±0.504% | 79.67%±0.809% | 78.14%±2.486% | 81.32%±1.901% | 81.09%±1.005% | 80.16%±1.353% | 82.33%±2.134% |
| 5 | 80.56%±1.149% | 83.85%±0.750% | 81.87%±0.638% | 80.34%±2.339% | 83.31%±1.529% | 83.37%±1.225% | 82.94%±0.830% | 84.19%±1.940% |
| 6 | 81.98%±1.334% | 85.61%±0.624% | 83.56%±0.541% | 82.32%±1.592% | 84.94%±0.858% | 85.19%±0.993% | 83.69%±0.975% | 85.80%±1.498% |
| 7 | 83.00%±1.048% | 86.62%±0.607% | 84.94%±0.079% | 83.98%±1.394% | 85.97%±1.179% | 86.31%±0.977% | 85.15%±0.760% | 86.75%±1.426% |
| 8 | 83.59%±0.945% | 87.57%±0.625% | 85.78%±0.068% | 85.26%±1.431% | 87.13%±0.679% | 87.55%±0.831% | 86.61%±0.478% | 87.91%±1.264% |
| 9 | 84.42%±0.744% | 88.23%±0.600% | 87.11%±0.437% | 86.18%±0.886% | 87.87%±0.598% | 87.89%±0.780% | 87.29%±0.441% | 88.52%±1.240% |

## Table 7: SVHN, VGG, Query Batch Size:1000, Initial Set Size:500

|   | RANDOM | MARG | ENTROPY | CORESET | CONF | ALBL | BADGE | *dynamicAL* |
|---|--------|------|---------|---------|------|------|-------|-------------|
| 0 | 25.90%±3.479% | 26.20%±5.409% | 26.85%±4.403% | 26.18%±6.853% | 27.21%±8.721% | 26.60%±4.688% | 26.88%±6.248% | 26.43%±8.047% |
| 1 | 70.26%±3.154% | 69.06%±3.646% | 68.72%±2.156% | 68.46%±1.111% | 69.85%±3.485% | 70.51%±3.487% | 70.09%±2.690% | 70.04%±1.650% |
| 2 | 77.91%±1.061% | 78.24%±2.237% | 78.56%±0.492% | 77.66%±1.784% | 78.89%±2.809% | 78.14%±1.494% | 78.67%±1.799% | 78.86%±1.710% |
| 3 | 81.25%±0.812% | 83.68%±1.657% | 82.83%±0.527% | 82.34%±1.461% | 83.75%±2.165% | 83.50%±1.669% | 83.07%±1.334% | 83.11%±1.269% |
| 4 | 83.63%±0.746% | 86.12%±1.251% | 85.80%±0.744% | 85.34%±1.126% | 85.91%±1.128% | 86.18%±0.979% | 86.50%±1.087% | 85.70%±1.179% |
| 5 | 85.17%±0.870% | 88.04%±1.022% | 87.66%±0.683% | 87.19%±0.928% | 87.61%±1.044% | 87.65%±1.031% | 88.03%±0.742% | 87.46%±1.054% |
| 6 | 86.06%±0.822% | 89.13%±0.712% | 88.96%±0.395% | 88.65%±0.505% | 88.90%±0.845% | 88.93%±0.809% | 88.41%±0.783% | 88.89%±1.274% |
| 7 | 87.30%±0.948% | 90.36%±0.532% | 90.00%±0.257% | 89.65%±0.486% | 90.18%±0.706% | 89.83%±0.747% | 90.53%±0.495% | 90.09%±1.149% |
| 8 | 87.69%±0.890% | 90.95%±0.375% | 90.67%±0.385% | 90.15%±0.410% | 90.96%±0.677% | 90.75%±0.567% | 91.25%±0.432% | 90.95%±0.782% |
| 9 | 88.28%±0.723% | 91.59%±0.417% | 91.25%±0.353% | 90.64%±0.311% | 91.66%±0.755% | 91.41%±0.665% | 91.76%±0.367% | 91.67%±0.840% |

Table 8: Caltech101, ResNet, Query Batch Size:500, Initial Set Size:500

| | RANDOM | MARG | ENTROPY | CORESET | CONF | ALBL | *dynamicAL* |
|---|---|---|---|---|---|---|---|
| 0 | 21.59%±1.431% | 21.98%±1.688% | 21.49%±1.681% | 21.39%±1.738% | 21.98%±1.459% | 21.48%±1.828% | 21.38%±1.323% |
| 1 | 25.84%±1.112% | 28.42%±1.677% | 27.43%±0.760% | 28.61%±1.224% | 27.25%±1.155% | 28.02%±1.515% | 29.34%±2.111% |
| 2 | 34.94%±0.635% | 34.76%±1.745% | 32.94%±1.224% | 35.37%±1.561% | 32.85%±0.849% | 35.86%±1.012% | 36.14%±1.234% |
| 3 | 37.34%±1.088% | 39.70%±1.328% | 36.36%±0.636% | 40.81%±1.005% | 37.52%±1.250% | 40.20%±1.091% | 41.19%±0.789% |
| 4 | 43.87%±0.867% | 43.26%±0.612% | 40.12%±0.805% | 45.38%±0.508% | 41.82%±1.104% | 45.27%±0.725% | 46.11%±1.138% |
| 5 | 45.45%±1.672% | 46.25%±1.562% | 43.24%±1.617% | 47.81%±1.683% | 44.60%±1.295% | 48.35%±1.729% | 49.42%±1.298% |
| 6 | 47.60%±1.383% | 49.20%±1.310% | 45.71%±1.047% | 50.60%±1.596% | 46.74%±0.760% | 51.20%±1.466% | 52.31%±1.739% |
| 7 | 49.97%±0.530% | 51.40%±1.571% | 48.19%±0.928% | 52.80%±1.887% | 49.19%±0.885% | 53.90%±1.166% | 55.03%±1.098% |
| 8 | 52.06%±1.476% | 53.56%±1.044% | 50.81%±0.943% | 55.31%±1.105% | 51.99%±1.383% | 56.22%±0.838% | 56.92%±1.153% |
| 9 | 54.04%±0.898% | 55.92%±0.496% | 53.05%±0.554% | 56.93%±0.691% | 54.96%±0.981% | 57.99%±0.805% | 58.81%±1.040% |

Table 9: Caltech101, ResNet, Query Batch Size:1000, Initial Set Size:500

| | RANDOM | MARG | ENTROPY | CORESET | CONF | ALBL | *dynamicAL* |
|---|---|---|---|---|---|---|---|
| 0 | 22.13%±1.050% | 22.05%±1.011% | 21.83%±0.725% | 20.98%±0.631% | 22.03%±1.364% | 22.05%±0.633% | 21.42%±1.735% |
| 1 | 33.91%±1.330% | 33.80%±1.002% | 31.98%±1.000% | 33.40%±0.962% | 32.43%±0.895% | 33.66%±2.174% | 33.83%±1.438% |
| 2 | 42.08%±0.560% | 41.22%±0.730% | 39.23%±0.981% | 43.24%±0.960% | 40.05%±0.988% | 43.28%±2.360% | 43.27%±2.280% |
| 3 | 47.43%±0.700% | 47.16%±0.659% | 46.26%±0.968% | 50.51%±0.706% | 47.87%±0.698% | 50.10%±2.082% | 50.43%±1.634% |
| 4 | 52.77%±0.980% | 54.52%±1.288% | 54.11%±1.347% | 56.15%±1.284% | 53.76%±1.196% | 56.96%±1.733% | 57.52%±1.189% |

## E.4  Maximum Mean Discrepancy for Multiple Rounds

As shown in Figure 1, the MMD term is much smaller than the $\mathcal{B}$ at the first query round. To better understand the relation between MMD and $\mathcal{B}$ for multiple query setting, we measure the MMD/$\mathcal{B}$ for $R \geq 2$. As shown in Figure 9, $\mathcal{B}$ is much larger than MMD even multiple query rounds. Besides, we notice that, for the first round, the larger query batch always leads to larger MMD/$\mathcal{B}$, because the sampling bias introduced by the query policy will be amplified by using large batch size.

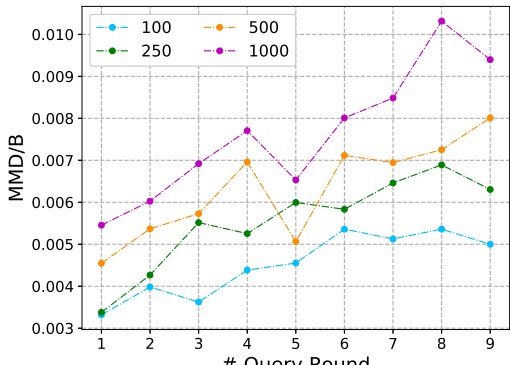

Figure 9: MMD/$\mathcal{B}$ for larger query round.

Furthermore, we measure the MMD/$\mathcal{B}$ with a constant total budget size but different query rounds. The result is shown in Table 10. As our expectation, spending the total query budget in one query round will induce the largest MMD/$\mathcal{B}$. And, with more query rounds, the MMD/$\mathcal{B}$ will be lower.

Table 10: MMD/$\mathcal{B}$ under constant budget size.

| SETTING | $R = 10, b = 100$ | $R = 4, b = 250$ | $R = 2, b = 500$ | $R = 1, b = 1000$ |
|---|---|---|---|---|
| MMD/$\mathcal{B}$ | 0.004999 | 0.005253 | 0.005367 | 0.005455 |

## E.5  Performance under the Re-initialization Setting

To study the effectiveness of *dynamicAL* under the re-initialization setting, we compare *dynamicAL* with the strong baseline involving the re-initialization trick in its algorithm design, e.g., Coreset [60].

Following [11], we query samples when training accuracy is greater than $99\%$ and the results are summarized in Table 11 and 12. The results show that *dynamicAL* can still be better than or competitive with the commonly used active learning methods. We notice that the improvement in the non-retraining setting is more significant. This is as our expectation. The dynamic analysis (Equation (8)), that *dynamicAL* is based on, considers the change of dynamics according to the model's current parameters. The re-initialization trick will not only causes the computational overhead of retraining, but also makes *dynamicAL* deviate from the analysis (Section 4).

## E.6  Performance with large query rounds

We provide the experiments with $b = 500, r = 15$ on Caltech101 data set with ResNet18 as the backbone. We ignore the BACKGROUND Google label and then we have 8677 images in total. At the last round, we run out of all images in the pool. As shown in the Figure 10, our method consistently outperforms those baselines. Note, due to the non-retraining setting, the model will have different performance even if all the samples are used for the training.

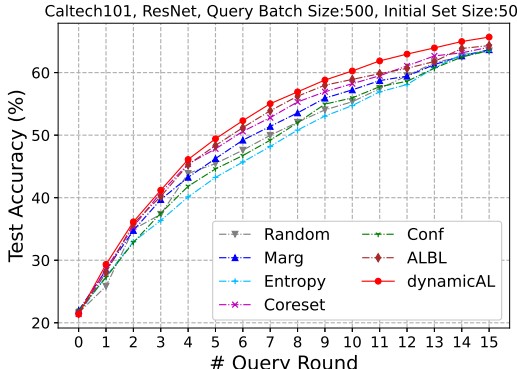

Figure 10: The evaluation result with larger query round on Caltech101.

Table 11: CIFAR10, ResNet, Query Batch Size 500, Initial Set Size 500.

| #ROUND | RANDOM | CORESET | *dynamicAL* |
|---|---|---|---|
| 0 | 30.80±1.81 | 30.77±0.92 | 30.94±2.17 |
| 1 | 35.80±1.52 | 36.62±2.10 | 36.47±0.13 |
| 2 | 42.91±1.75 | 43.16±1.79 | 42.74±2.44 |
| 3 | 43.76±0.65 | 44.35±2.25 | 46.43±1.07 |
| 4 | 47.03±1.19 | 48.74±1.94 | 49.38±1.80 |
| 5 | 49.16±1.77 | 50.20±1.25 | 51.61±1.09 |
| 6 | 52.43±1.33 | 53.44±1.37 | 54.33±1.76 |
| 7 | 52.81±1.55 | 53.89±0.78 | 54.59±1.04 |
| 8 | 54.56±0.23 | 57.12±1.11 | 57.50±1.28 |
| 9 | 58.08±1.48 | 59.62±1.50 | 60.35±1.80 |

Table 12: SVHN, VGG, Query Batch Size 500, Initial Set Size 500.

| #ROUND | RANDOM | CORESET | *dynamicAL* |
|--------|--------|---------|-------------|
| 0 | 52.68±1.97 | 52.74±6.16 | 52.59±3.73 |
| 1 | 67.64±1.99 | 68.08±3.61 | 66.48±4.10 |
| 2 | 73.46±1.51 | 74.93±1.44 | 74.34±2.22 |
| 3 | 77.30±1.08 | 76.49±2.08 | 76.73±2.65 |
| 4 | 79.27±0.78 | 79.33±0.72 | 80.19±0.78 |
| 5 | 79.97±1.28 | 82.09±1.08 | 82.08±1.39 |
| 6 | 83.97±0.42 | 82.30±0.33 | 83.80±1.30 |
| 7 | 83.44±0.57 | 83.29±1.11 | 84.85±1.12 |
| 8 | 86.24±0.52 | 84.72±0.52 | 86.59±1.25 |
| 9 | 85.75±1.23 | 85.62±0.55 | 86.57±0.74 |

# F    Discussion

**Limitation and Future Work.**    In the work, we study the connection between generalization performance and the training dynamics under the NTK regime. Although the relation between training dynamics and generalization performance has been verified by our experiments, the theoretical analysis of the relation out of the NTK regime still needs study. Besides, in the experiments, we mainly focus on the classification problem. Whether the proposed method is effective for the regression problem is under-explored. We would like to leave the study of the previously mentioned two problems in the future work.

**NTK Analysis for the Design of Practical Method.**    Although some works [64, 65] discussed that the NTK assumption is hard to be strictly satisfied in some real-world models, we notice that some recent works have shown that the high-level conclusions derived based on NTK is insightful and useful for the design of practical models. Some of their applications can achieve SOTA. For example, Park et al. [66] used the NTK to predict the generalization performance of architectures in the application of Neural Architecture Search (NAS). Chen et al. [67] used the condition number of NTK to predict a model's trainability. Chen et al. [68] also used the NTK to evaluate the trainability of several ImageNet models, such as ResNet. Deshpande et al. [69] used the NTK for model selection in the fine-tuning of pre-trained models on a target task. In our work, the empirical results in Figure 3 and Appendix.E also show the effectiveness of the high-level conclusions derived from the theory still hold.

**Social Impacts.**    In this work, we study the connection between the generalization performance and the training dynamics and try to bridge the gap between the theoretic findings of deep neural networks and deep active learning applications. We hope our work would inspire more attempts on the design of deep active learning algorithms with theoretical justification, which might have positive social impacts. We do not foresee any form of negative social impact induced by our work.

**License Privacy Information.**    We use the commonly used datasets, CIFAR10[4], SVHN[5], Caltech101[6] in the experiments. Those datasets follow the MIT, CC0 1.0, CC BY 4.0 License respectively and are publicly accessible. No privacy information is included in those datasets.

---

[4] https://www.cs.toronto.edu/~kriz/cifar.html
[5] http://ufldl.stanford.edu/housenumbers/
[6] https://data.caltech.edu/records/20086