# OpenReview forum: "Deep Active Learning by Leveraging Training Dynamics"
_NeurIPS.cc/2022/Conference — NeurIPS 2022 Accept_

### Official Review · Reviewer_7LZg · 2022-07-11

**Rating:** 5
**Confidence:** 2
**Soundness:** 3 good
**Presentation:** 2 fair
**Contribution:** 3 good

**Summary:**

The authors prove that convergence speed of training and generalization are positively correlated. Based on this observation, the authors proposed a novel active learning algorithm (dynamicAL), which maximizes training dynamics. DynamicAL is theoretically motivated and generalizes existing methods such as uncertainty sampling methods and influence function methods.


**Questions:**

Can you compare the proposed method with other baseline methods for rounds > 10?

Can you discuss the tightness of the bound in the relationships between convergence and generalization?


**Limitations:**

Yes

**Strengths And Weaknesses:**

Originality:

The paper examines the training dynamics of Neural Tangent Kernel, which is suitable for theoretical analysis. The authors proved that convergence speed and generalization are positively correlated.
The authors proposed a new objective for an active learning algorithm: optimize the speed of training convergence.

Quality:
The results are novel and interesting. However, the quality of the bound is not clear. For example, the authors introduce the notion of label alignment and show that label alignment is correlated with convergence and generalization. Based on this correlation, the authors conclude that convergence and generalization are positively correlated. While it is logical, I believe it oversimplifies the dependency between convergence and generalization given that the bound is an approximation and the quality of the bound was not assessed.
Experimental results can be improved by measuring performance for all rounds (> 10 rounds until the training set is exhausted). Currently, the authors evaluate and compared methods for less than 10 query rounds. It is not clear if the proposed method outperforms the baselines in all rounds.
The improvement is rather small (often less than 1%).

Clarity:
In general, the paper is easy to read. However, its flow and presentation can be improved. For example, the authors first present the method and later present a theoretical analysis of “Train faster → generalize better”. I suggest that some of the theoretical results can be presented together with the method to improve the presentation quality. Other, unnecessary lemmas should be moved to the appendix. Then, the authors can expand the theoretical section and discussion of related work sections.

Significance:
The authors’ idea of optimizing for convergence speed is interesting and of potential significance, as it can be explored by a wider community in other fields. However, the authors should improve the presentation and clarify of the paper, while also updating experimental results.

---

> ### Author Response · Authors · 2022-08-02
> **Author Response to Reviewer 7LZg**
>
> 1. Can you compare the proposed method with other baseline methods for rounds > 10?
> * We provide the experiments with $b=500, r=15$ on Caltech101 data set with ResNet18 as the backbone in Appendix E.6. At the last round, we run out of all images in the pool. As shown in the Figure 10, our method consistently outperforms those baselines.
>
> 2. Can you discuss the tightness of the bound in the relationships between convergence and generalization?
> * By looking at Eq (19), the lower bound is due to the HM-GM-AM-QM inequalities which state the relationship between the harmonic mean, geometric mean, arithmetic mean, and quadratic mean. The tightness of the lower bound depends on the value distribution of $\lambda_i(v_i^{\top}Y)^2$. If $\lambda_i(v_i^{\top}Y)^2$ is close to each other, then the generalization performance is close to the lower bound. On the other hand, if the difference among $\lambda_i(v_i^{\top}Y)^2$ is significant, then the upper bound should be more close to the generalization performance, because the upper bound has $\frac{\lambda_{\max}}{\lambda_{\min}}$ to account for the size of the region of the eigenvalue distribution.

---

### Official Review · Reviewer_Sva2 · 2022-07-11

**Rating:** 7
**Confidence:** 4
**Soundness:** 3 good
**Presentation:** 3 good
**Contribution:** 3 good

**Summary:**

In this paper, the authors address the problem of active learning in the setting where the learning model is over-parameterized with respect to the data.  They note that theories of active learning are based upon assumptions that hold for hypothesis classes with lower parameterizations may not hold in the over-parameterized setting, and propose to use newer tools from generalization theory to theoretically ground active learning for deep neural networks.  They propose to use tools for studying the relationship between training dynamics and generalization performance (such as the NTK) as the basis for new deep active learning heuristics to select unlabeled data points in the pooled setting, eschewing traditional methods that select points that maximize either labeled data diversity or model uncertainty.   They demonstrate that selecting data points which maximize the rate at which models train (the rate of descent of the training loss as a function of the number of parameter updates) is positively correlated with a lower generalization bound, thus establishing it as a bona fide heuristic for the acquisition function.


**Questions:**

Lines 284-285: I would recommend trying this out on datasets with more severe class imbalances.  Each of CIFAR10, SVHN and Caltech 101 are rather well behaved and don’t exhibit any class imbalances, making them odd choices for active learning.  Perhaps this is why there does not seem to be much of a drastic difference in the results of Figure 3?

Line 162: The use of pseudo labeling in this paper connects this work to recent work on self-supervised learning.  I wonder if the authors have considered the implications of what current theories of self-supervised learning can do to inform this work, or what this work could do to transform a self-supervised learning method into a few-shot method that would preform better?


**Limitations:**

Yes, they have.

**Strengths And Weaknesses:**

Lines 49-52: Will definitely have to ensure that TD can be computed or well-estimated quickly, lest this be impractical

Line 100: $K^{i,j}(x,x’)$ is the inner product of the gradient of the $i$-th class probability and the gradient of the $j$-th class probability evaluated at {x} and {x’} respectively.  I think?  It's not totally clear, and having to flip back to the notation section much higher up is not ideal.

Line 122: Similar criticism here.  This section includes a lot of notation, it would help the reader here to better understand equation (8) to recall that $K^{ij}(X,X)$ is the empirical NTK.

Line 150: This is a common issue in active learning.  This can bias the selection of points, since the point-wise contribution to G(S) is not necessarily additive.  See work by [Farquahr et al 2021](https://arxiv.org/abs/2101.11665) for an example in the case of using uncertainty to acquire labels.  A test to see how vulnerable G(S) is to this phenomenon would be to run selection with $b = 1$, and then with $b$ increasing in various intervals, to see how frequently the single points acquired with $b = 1$ appear in the  set of $b$ points acquired in a single batch selection.

Line 154: I presume that $\mathcal{K}^{ij}$ is the empirical NTK, but it would help the reader to specify what it represents, as it’s not super clear by the derivations from lines 149 - 155.

Lines 176-178: While using an IF derived acquisition function makes some sense (in the sense that high influence points will by identification provoke large magnitude changes in the model parameters), it’s not clear that it’s a better or more valuable way to acquire points.

Line 194: “of” missing here

Lines 217-219: In theorem 1, the factor $\nu$ will change at $t -> \infty$, since it depends on the reciprocal of $n^2$.  Is this accounted for in the proof?

Line 260: I would reformulate this sentence here, since evaluating one experiment for two query rounds on one network and dataset does *not* do much to convince the reader that Lemma 2 holds.  I think the authors should broaden the datasets they use to establish the validity across other datasets, especially with different class imbalances.

Line 263: This figure is nice, but it lacks some context.  How good is the correlation between the training dynamics proxy and alignment?  What would help readers understand this better would be a comparison of different acquisition functions and how they correlate with alignment.  This correlation argument would become much stronger.

Section 5: There are two key points missing in the empirical analysis of dynamicAL versus other acquisition functions.

The first is an analysis of the degree of overlap between the points selected by dynamicAL and other methods; how many of the same points are chosen, and does the membership of the sets of acquired points converge over time, or do they diverge as successive rounds of acquisition take place?

The second is that in both figure 3 and 5 we see only the means but not the variances of each set of retraining experiments.  I suspect that there are two components contributing to the performance of each algorithm: first is the selection of the points acquired, as well as the smaller differences in training dynamics for each network due to initialization, batch order in which the data is presented to the models, or even GPU noise.  It would be more convincing to see error bars, so that we can gauge how much of an effect the selection of points has on the accuracy.

---

> ### Author Response · Authors · 2022-08-02
> **Author Response to Reviewer Sva2**
>
> 1. To help the readers better understand a) provide more descriptions for the empirical NTK in line 100; b) recall that $\mathcal{K}^{i j}$ is the empirical NTK in lines 122, 154.
> * Thank you for your constructive comments! As suggested, we clarify the meaning of the notation $\mathcal{K}^{i j}$ in line 100 and recall the meaning of this notation in lines 122, 156.
>
>
> 2. Figure 2 is nice, but it lacks some context. How good is the correlation between the training dynamics proxy and alignment? What would help readers understand this better would be a comparison of different acquisition functions and how they correlate with alignment.
> * Thanks for the suggestion. We provide more description and explanation for Figure 2 to help readers understand the correlation between the training dynamics and alignment. But it can be challenging to bridge the acquisition functions of other active learning methods to alignment, which is related to the generalization bound (Theorem 2). Actually, the connection with the theoretical analysis is a distinct advantage of our method.
>
>
> 3. The subset approximation introduced in line 150 can bias the selection of points, since the point-wise contribution to $G(S)$ is not necessarily additive. See work by Farquahr et al.[1] for an example in the case of using uncertainty to acquire labels.
> * To analyze the feasibility of this approximation, we empirically measure the expectation of the Approximation Ratio (Eq (14)) on three data sets with three different neural networks in Figure 4 which show that the approximated result is close to the original value. Then, the samples selected based on the approximated value will not be largely different from the samples selected with the original one.
> *  The paper [1] pointed out by the reviewer studies the influence of statistical bias commonly existing in active learning methods and finds the bias can be actively helpful when training with neural networks. We acknowledge that the point-wise contribution is also not strictly additive in our case and studying the influence of statistical bias for our method is interesting, but it is out of the scope of this work. We will continue the study of the statistical bias for the generalization performance of our method in our future work.
>
>
> 4. (Lines 49-52) Ensure that TD can be computed or well-estimated quickly.
> * The training dynamics will not be directly computed in our method. Instead, we compute the change of training dynamics, as shown in Eq (12). Furthermore, to accelerate the computation of the acquisition function of Eq (12), we introduce the subset approximation which leads to the criterion employed to select samples Eq (15). We provide additional discussion of the computational requirement of Eq (15) in Appendix C of the revision.
>
> [1] Farquhar, Sebastian, Yarin Gal, and Tom Rainforth. "On Statistical Bias In Active Learning: How and When to Fix It." International Conference on Learning Representations. 2020.

---

> > ### Author Response · Authors · 2022-08-02
> > **Author Response to Reviewer Sva2 (cont.)**
> >
> > 5. I would recommend trying this out on datasets with more severe class imbalances.
> > * Thanks for your suggestion. In this work, we follow the previous works [6, 7, 8] and adopt the commonly used data sets for active learning. On those balanced data sets, we verify the correctness of the theoretical analysis and the effectiveness of the proposed method. The study of active learning with imbalanced data sets is important and interesting. The common setting of class imbalance is to train the model with class imbalanced data and test on balanced data [10, 11]. However, the distribution discrepancy between training and test data will complicate the generalization analysis (Theorem 2). Note that the generalization analysis under the distribution shift problem is an actively studied independent field [9] out of the scope of this paper. For this work, we want to make the experiments close to the analysis and would like to leave the exploration on this setting in future work. The other setting is to train and test the model on the data from the same imbalanced distribution. The data set Caltech101 we use in the experiment is in this setting, and we can observe better performance improvement (Figure 10 in Appendix E.6). We would like to try dynamicAL on this imbalanced setting on other more imbalanced data sets next.
> >
> >
> > 6. What current theories of self-supervised learning can do to inform this work, or what this work could do to transform a self-supervised learning method into a few-shot method that would perform better?
> > * Several works are trying to pretrain the network with a self-supervised learning (SSL) method before the active learning method [2, 3]. The formal analysis with the self-supervised learning theory [4,5] is an interesting and important topic to explain the empirical success of using SSL to warmup for active learning.  In [2], the proposed method relying on the pseudo-labeling, enjoys the benefit introduced by pretraining. For our work, the SSL warmup might empirically bring benefit to our methods, although this is beyond the scope of the current method.
> >
> > 7. Lines 217-219: In theorem 1, the factor $v$ will change at $t \rightarrow \infty$, since it depends on the reciprocal of $1/n^2$. Is this accounted for in the proof?
> >
> > * Our theoretical result is based on the fact that neural tangent kernel stays constant during gradient descent training. In particular, $\Theta= V^{\top} \Lambda V$, where $V= $ \{ $ v_{i} $ \}$ _{i=1}^n$. Because neural tangent kernel stays constant, $v_i$ will keep unchanged during gradient descent update.
> >
> >
> >
> > [2] Gudovskiy, Denis, et al. "Deep active learning for biased datasets via fisher kernel self-supervision." Proceedings of the IEEE/CVF Conference on Computer Vision and Pattern Recognition. 2020.
> >
> > [3] Bengar, Javad Zolfaghari, et al. "Reducing label effort: Self-supervised meets active learning." Proceedings of the IEEE/CVF International Conference on Computer Vision. 2021.
> >
> > [4] Arora, Sanjeev, et al. "A theoretical analysis of contrastive unsupervised representation learning." arXiv preprint arXiv:1902.09229 (2019).
> >
> > [5] HaoChen, Jeff Z., et al. "Provable guarantees for self-supervised deep learning with spectral contrastive loss." Advances in Neural Information Processing Systems 34 (2021): 5000-5011.
> >
> > [6] Ash, Jordan T., et al. "Deep batch active learning by diverse, uncertain gradient lower bounds." arXiv preprint arXiv:1906.03671 (2019).
> >
> > [7] Liu, Zhuoming, et al. "Influence selection for active learning." Proceedings of the IEEE/CVF International Conference on Computer Vision. 2021.
> >
> > [8] Zhang, Beichen, et al. "State-relabeling adversarial active learning." Proceedings of the IEEE/CVF conference on computer vision and pattern recognition. 2020.
> >
> > [9] Chuang, Ching-Yao, Antonio Torralba, and Stefanie Jegelka. "Estimating generalization under distribution shifts via domain-invariant representations." arXiv preprint arXiv:2007.03511 (2020).
> >
> > [10] Aggarwal, Umang, Adrian Popescu, and Céline Hudelot. "Active learning for imbalanced datasets." Proceedings of the IEEE/CVF Winter Conference on Applications of Computer Vision. 2020.
> >
> > [11] Bengar, Javad Zolfaghari, et al. "Class-Balanced Active Learning for Image Classification." Proceedings of the IEEE/CVF Winter Conference on Applications of Computer Vision. 2022.

---

> > > ### Comment · Reviewer_Sva2 · 2022-08-08
> > > **Thanks!**
> > >
> > > Thanks for an invigorating discussion, this paper was a pleasure to review.  I hope that others read it with as much enthusiasm as I did :)

---

> > > > ### Author Response · Authors · 2022-08-09
> > > > **Thank you very much!**
> > > >
> > > > Thank you very much for the positive comments. We really appreciate all the discussions and suggestions!
> > > >
> > > >
> > > >
> > > > Best,
> > > >
> > > > Authors.

---

> > ### Comment · Reviewer_Sva2 · 2022-08-08
> > **Reviewer response to authors**
> >
> > > But it can be challenging to bridge the acquisition functions of other active learning methods to alignment, which is related to the generalization bound (Theorem 2). Actually, the connection with the theoretical analysis is a distinct advantage of our method.
> >
> > Sorry, I should have been more clear here.  The experiment I was suggesting was to run two acquisition functions per budgeted time step; one based on calculating alignment, the other based on some heuristic acquisition function against which you compare.  That way, you can report not only the difference in points that each acquisition function suggests would lead to the greatest expected improvement, but also you can learn more about alignment by looking at the difference in alignment distributions between your method and competing methods.  In this way, you might learn more about how alignment contrasts with existing methods.  I still think it would be a helpful guide to readers, but I won't insist on piling on more work for work's sake.
> >
> > > The paper [1] pointed out by the reviewer studies the influence of statistical bias commonly existing in active learning methods and finds the bias can be actively helpful when training with neural networks. We acknowledge that the point-wise contribution is also not strictly additive in our case and studying the influence of statistical bias for our method is interesting, but it is out of the scope of this work. We will continue the study of the statistical bias for the generalization performance of our method in our future work.
> >
> > That's a fair distinction, and I appreciate that while important, you cannot fit all the work that's interesting into one paper.  I hope you'll follow up on it in future work.

---

### Official Review · Reviewer_yzw6 · 2022-07-11

**Rating:** 7
**Confidence:** 4
**Soundness:** 3 good
**Presentation:** 4 excellent
**Contribution:** 3 good

**Summary:**

1. The paper describes DynamicAL, an algorithm for active learning based on NTK-based analysis of training dynamics
2. The algorithm is based on the notion that fast training results in better generalization. The authors describe "alignment" as a measure of convergence rate and show that this term affects the generalization bounds derived in the NTK regime.
3. The authors then show that the proxy used in DynamicAL ("training dynamics") correlates with alignment and therefore could be used as a proxy
4. The paper propose a greedy variant of the proposed algorithm and show that it well approximates the non-greedy variant, while being significantly faster
5. The paper shows that the resulting algorithm outperforms existing baselines on a few models/datasets

**Questions:**

1. What is the runtime of the algorithm? It appears that calculating equation 15 would scale linearly with |S|, which seems quite undesirable

2. Does the algorithm suffer from clustering due to the greedy approximation? As an example, suppose that every single item in the dataset is duplicated 10 times and the query size is 10. Will the algorithm not pick the exact same image 10 times? (see the distinction between BALD and BatchBALD, for example)

3. Is the experiment in section 4.3 calculating G using pseudo-labels or ground truth labels? It would be interesting to see the effect of changing pseudo-labels to ground-truth labels on the results in section 4.3

4. While section 4.3 shows that G (training dynamics) correlates with A (alignment), it would useful to see how well G correlates to the final generalization performance, as this is true objective of the active learning algorithm.

**Limitations:**

See the questions and strength/weaknesses section

**Strengths And Weaknesses:**

Strengths:
1. Good presentation, well structured
2. Well motivated, main claims of the papers are well justified from theoretical point of view, and verified empirically (section 4)
3. Resulting algorithm is simple to implement and could be considered generalizations of existing methods (as mentioned in section 3.3)

Weaknesses:
1. Derivation and algorithm based heavily on NTK theory, but practical sized networks have been shown to deviate far from this regime (see [1, 2], for example). This paper does not address this issue. At least some discussion of it would be useful
2. Unclear computational requirements compared to other methods or runtime analysis (this part, while possible to interpret based on the algorithm description, could be stated more explicitly)
3. Improvements over baselines seem rather marginal (although this is somewhat expected in a somewhat saturated field)

[1] - https://arxiv.org/abs/2007.15801
[2] - https://arxiv.org/abs/2010.15110

---

> ### Author Response · Authors · 2022-08-02
> **Author Response to Reviewer yzw6 (1/3)**
>
> Thank you very much for the recognition of our work!  We’d like to address your questions as follows.
>
> 1. Derivation and algorithm based heavily on NTK theory, but practical sized networks have been shown to deviate far from this regime (see [1, 2], for example). This paper does not address this issue. At least some discussion of it would be useful.
> * Thanks for your suggestion. We provide more discussion of the practical sized networks in Appendix F of the revision. Although [1, 2] mentioned that the NTK assumption is hard to be strictly satisfied in some real-world models, we notice that some recent works have shown the high-level conclusions derived based on the NTK analysis are insightful and useful to guide the design of the practical methods. Some of their applications can achieve SOTA. For example,
>     * Park et al. [3] used the NTK to predict the generalization performance of architectures in the application of Neural Architecture Search (NAS).
>     * Chen et al. [4] used the condition number of NTK to predict a model’s trainability.
>     * Chen et al. [5] also used the NTK to evaluate the trainability of several ImageNet models, such as ResNet.
>     * Deshpande et al. [6] used the NTK for model selection in the fine-tuning of pre-trained models on a target task.
>
>     In our work, the empirical results in Figure 3 and Appendix E also show that the high-level conclusion (Proposition 1) still holds.
>
>
> 2. It appears that calculating Eq (15) would scale linearly with |S|, which seems quite undesirable.
> * The runtime is almost the same with the increase of the size of set S. For the computation of Eq (15),
>
>     $\Delta(\{(x_u, \widehat{y}_u) \}| S) = $
>
>     $\|\nabla_{\theta} \ell (f({x}_u; \theta), \hat{{y}}_u)\|^2 $
>
>     $+ 2 \sum_{(x, y) \in S} \nabla_{\theta}\ell(f({x}_u; \theta), \hat{{y}}_u)^{\top}$
>
>     $ \nabla_{\theta} \ell (f({x}; \theta), {{y}})$
>
>     the second term can be rewritten as
>
>     $2 \nabla_{\theta}\ell(f({x}_u; \theta), \hat{{y}}_u)^{\top}  $
>
>     $\sum_{(x, y) \in S} \nabla_{\theta} \ell (f({x}; \theta), {{y}})$
>
>     The summation of gradient over the set $S$ needs to be computed only once and reused for each sample $(x_u, \hat{y}_u)$. Although the summation operation is linear with $|S|$, its computational overhead can be ignored, compared with the computation of gradient inner product over the unlabeled set. Furthermore, to help the reader better understand the computational requirements, we provide a discussion in Appendix C.2.
>
> [1] Lee, Jaehoon, et al. "Finite versus infinite neural networks: an empirical study." Advances in Neural Information Processing Systems 33 (2020): 15156-15172.
>
> [2] Fort, Stanislav, et al. "Deep learning versus kernel learning: an empirical study of loss landscape geometry and the time evolution of the neural tangent kernel." Advances in Neural Information Processing Systems 33 (2020): 5850-5861.
>
> [3] Daniel S Park, Jaehoon Lee, Daiyi Peng, Yuan Cao, and Jascha Sohl-Dickstein. Towards nngp guided neural architecture search. arXiv preprint arXiv:2011.06006, 2020.
>
> [4] Wuyang Chen, Xinyu Gong, and Zhangyang Wang. Neural architecture search on imagenet in four gpu hours: A theoretically inspired perspective. In International Conference on Learning Representations, 2021a.
>
> [5] Chen, Xiangning, Cho-Jui Hsieh, and Boqing Gong. "When Vision Transformers Outperform ResNets without Pretraining or Strong Data Augmentations." arXiv preprint arXiv:2106.01548 (2021).
>
> [6] Deshpande, A., Achille, A., Ravichandran, A., Li, H., Zancato, L., Fowlkes, C., Bhotika, R., Soatto, S. and Perona, P., 2021. A linearized framework and a new benchmark for model selection for fine-tuning. arXiv preprint arXiv:2102.00084.

---

> > ### Author Response · Authors · 2022-08-02
> > **Author Response to Reviewer yzw6 (2/3)**
> >
> > 3. Whether the proposed method is vulnerable to clustering.
> > * Theoretically, our method is not vulnerable to the clustering issue. The dynamicAL aims at selecting samples maximally increasing the training dynamics (Eq (13)). We can rewrite the original function in the following form with the gradient of each sample:
> >
> >     $Q^{\star} = argmax_{Q \subseteq U} \| \sum_{(x_u,y_u)\in Q} \nabla_{\theta} \ell(f(x_{u} ; \theta), {y}_{u})\|^{2}$
> >
> >     $+2 \sum_{(x, y) \in S} \nabla_{\theta} \ell(f(x_{u} ; \theta), {y}_{u})^{\top} $
> >
> >     $\nabla_{\theta} \ell(f(x ; \theta), y)$
> >
> >     The  optimal solution will not change by adding a constant $C$ into the acquisition function,
> >
> >     $Q^{\star} = argmax_{Q \subseteq U} \| \sum_{(x_u,y_u)\in Q} \nabla_{\theta} \ell(f(x_{u} ; \theta), {y}_{u})\|^{2}$
> >
> >     $+2 \sum_{(x, y) \in S} \nabla_{\theta} \ell(f(x_{u} ; \theta), {y}_{u})^{\top} $
> >
> >     $\nabla_{\theta} \ell(f(x ; \theta), y) + C$
> >
> >     Without loss of generality, let $C = \| \sum_{(x,y) \in S} \nabla_{\theta} \ell(f(x ; \theta), y)\|^{2}$, then we have,
> >
> >     $Q^{\star} = argmax_{Q \subseteq U} \|\mathbf{g}_{Q} $
> >
> >     $+ \mathbf{g}_{S} \|^{2}$
> >
> >     where $\mathbf{g}_{Q}$
> >
> >     $ = \sum_{(x_u,y_u)\in Q} \nabla_{\theta} \ell(f(x_{u} ; \theta), {y}_{u})$
> >
> >     and $\mathbf{g}_{S} $
> >
> >     $= \sum_{(x,y)\in S} \nabla_{\theta} \ell(f(x ; \theta), {y})$.
> >
> >     By assuming that outliers, which can cause extraordinarily large gradient norms, don't exist in the data set, we need to select a set from the candidate pool, where its composed gradient, $\mathbf{g}_Q$, has the same direction as $\mathbf{g}_S$. Back to your example, unless there exists a sample causing the gradient to be coincidentally the same as $\mathbf{g}_S$, the same image will not be selected 10 times.
> > * Empirically, we use the approximated acquisition function (Eq (15)). The important difference is that $\sum_{(x_u,y_u)\in Q} \|\nabla_{\theta} \ell(f(x_{u} ; \theta), {y}_{u})\|^2$ is used to replace the first term in the original form. And we verify the feasibility of the approximation over non-degenerate data sets. For the degenerate data sets, when the ratio of duplication $r \ll b$, where $b$ is the query batch size, the difference can be ignored. And dynamicAL will **not** suffer from the clustering issue. For example, if the data set is duplicated 10 times (as your example), the selected samples will almost remain the same with $b \in$ \{250, 500, 1,000\} as adopted in this work. We acknowledge that the approximation might cause an issue when the duplication ratio is much larger than the query batch, such as each image of the data set being duplicated 10,000 times, but this extreme case is out of the scope of this work. We explicitly give the non-degenerate assumption[7] in the revision to better clarify the scope of this work.
> >
> > [7] Allen-Zhu, Zeyuan, Yuanzhi Li, and Zhao Song. "A convergence theory for deep learning via over-parameterization." International Conference on Machine Learning. PMLR, 2019.

---

> > > ### Author Response · Authors · 2022-08-02
> > > **Author Response to Reviewer yzw6 (3/3)**
> > >
> > > 4. Is the experiment in section 4.3 calculating G using pseudo-labels or ground truth labels? It would be interesting to see the effect of changing pseudo-labels to ground-truth labels on the results in section 4.3.
> > > * In Figure 2, we compute $G(S \cup \overline{Q})$ in which $\overline{Q}$ is the corresponding data set with ground-truth labels (line 134). In the Appendix, we further provide the relationship between $\mathcal{A}(X \| X_Q, Y \| Y_Q)$ and $G(S \cup \widehat{Q})$, where $\widehat{Q}$ includes the pseudo-labels. Comparing Figure 2 and Figure 6, we find that the positive relationship between $\mathcal{A}$ and $G$ computed with ground-truth labels is slightly stronger than $G$ computed with pseudo-labels. The result is aligned with our expectations, because the extra noise is introduced in the pseudo-labels. However, the Kendall $\tau$ coefficient still achieves $0.46$ for $\mathcal{A}$ and $G$ computed with pseudo-labels. The result justifies the use of $G(S \cup \widehat{Q})$ as the acquisition function to query samples, which is verified by our empirical results (Figure.3 and Appendix E.5).
> > > * We refer the reviewer to Appendix D.2 of the revision for a more detailed discussion and the result.
> > >
> > >
> > >
> > > 5. It would useful to see how well G correlates to the final generalization performance.
> > > * We provide the correlation between $G$ and bound $B$ in the Appendix D.3 of the revision. Figure 7 shows that with the increase of $G$, $B$ will decrease. This empirical observation is aligned with our expectation, because Theorem 2 indicates that the alignment $A$ is inverse proportional to $B$ and Figure 2 tells us that the $G$ is proportional to $A$. We would like to refer the reviewer to the Appendix D.3 of the revision for a more detailed discussion and the result.

---

> > > > ### Comment · Reviewer_yzw6 · 2022-08-04
> > > > **Reviewer Response**
> > > >
> > > > Thanks for addressing the points brought up in the initial review. I wanted to ask again about the clustering problem with the greedy approximation. I understand that the non-greedy version of your approach does not suffer from clustering, but I was wondering about the greedy version specifically (eq. 12).
> > > >
> > > > If I understand correctly, the greedy acquisition function is:
> > > >
> > > > $Q_{greedy}^* = argmax_{Q \subseteq U} \sum_{\{x_i, y_i\} \in Q} \left( ||\nabla_\theta l(f_\theta(x_i), y_i) ||^2_2 + \nabla_\theta l(f_\theta(x_i), y_i) ^ T g_S\right)$
> > > >
> > > > Note this is different than what you provided in your response since
> > > >
> > > > $ \sum_{\{x_i, y_i\} \in Q} ||\nabla_\theta l(f_\theta(x_i), y_i) ||^2_2 \neq || g_Q||^2_2 $
> > > >
> > > > so it looks like the greedy variant of the algorithm will target the top |b| which match the gradient of the full training set **individually**, as opposed to matching the gradient when combined together. This seems like it would be susceptible to clustering.

---

> > > > > ### Author Response · Authors · 2022-08-05
> > > > > **Response to Reviewer yzw6**
> > > > >
> > > > >
> > > > > Thanks for your reply!
> > > > >
> > > > > As mentioned, the non-greedy version does not suffer from clustering. The greedy version is introduced for the fast approximation of the non-greedy one  (as described in Section 3.2, Line 147). Therefore, a natural question is: *what is the quality of the approximation?* To study the quality and feasibility of the approximation, we introduce the term Approximation Ratio (Eq(14)). When the approximation ratio is close to 1, then it means that the approximated value is close to the original one. As shown in Figure.4, under 6 different settings, the approximation ratio is closed to 1, which indicates that for real-world datasets (in general, the non-degenerated data), the approximation is good enough. Then the greedy version doesn’t suffer from the clustering issue.
> > > > >
> > > > > On the other hand, for the case mentioned in the initial post, the approximation ratio is closed to 1 when the times of duplication are much smaller than the query batch size. As we discussed in the initial reply, when the duplication ratio is larger than the query batch, the approximation will deviate from the original value, in which the clustering issue may happen.
> > > > >
> > > > > We acknowledge that the studies of when the employed approximation is invalid and whether we can have a better approximation are valuable. But, the extreme case doesn’t belong to the non-degenerated data as we assumed in Section.2, and thus it is out of the scope of this work. We would like to leave the exploration of it to future work.

---

> > > > > > ### Comment · Reviewer_yzw6 · 2022-08-05
> > > > > > **Response**
> > > > > >
> > > > > > Thanks for the reply. I will be maintaining my recommendation for acceptance (7).

---

### Official Review · Reviewer_GLZg · 2022-07-12

**Rating:** 6
**Confidence:** 3
**Soundness:** 3 good
**Presentation:** 3 good
**Contribution:** 3 good

**Summary:**

The paper proposes an active learning method for deep learning (named dynamicAL) that selects data points which maximize the training dynamics. The paper proves a relationship between the convergence speed of training and generalization and shows experimentally that dynamicAL can outperform other popular active learning baselines.

**Questions:**

- Can you prove formal results similar to Theorem 1 and 2 for the active learning setting in Section 4.2 and 4.3 ?

**Limitations:**

There is no potential negative societal impact.

**Strengths And Weaknesses:**

Strengths:
- The method proposed in the paper is novel and works well compared to other baselines.
- The experiments are extensive.
- The paper is well-written.

Weaknesses:
- The main theoretical results in Section 4.1 (Theorem 1 and 2) are only for iid data, which do not hold for the active learning setting. Although Section 4.2 gives some discussions for the active learning setting, the paper does not prove any formal results for the setting.
- Some notations can be simplified to make the formulas cleaner; for example in Eq 13, the subscript u for x, y, etc. can be dropped without affecting the meaning of the equation. The same is also true for other equations such as 12, 14, etc.

---

> ### Author Response · Authors · 2022-08-02
> **Author Response to Reviewer GLZg**
>
> Thank you for your constructive comments!
>
> 1. Formal results similar to Theorems 1 and 2 in the active learning setting.
> * Because Lemma 1 does not require the training set to be i.i.d., we can directly apply Theorem 1 to the active learning setting.
> * As for Theorem 2 (generalization upper bound), we can establish a generalization upper bound for the active learning setting by plugging Eq (20) into Eq (18). And we show the result here
>
>     $\mathcal{L}_p \le \sqrt{\frac{2 \textrm{Tr}[Y^{\top}\Theta^{-1}(X,X)Y]}{n}}$
>
>     $ +\textrm{MMD}(S_0,S,\mathcal{H}_{\Theta}) $
>
>     $ + O(\sqrt{\frac{\log\frac{n}{\lambda_{0}\delta}}{n}}) +O(\sqrt{\frac{C\log(1/\delta)}{n}})$.
>
>     Then the leading term in the generalization bound will be:
>
>     $\mathcal{B}'= \sqrt{\frac{2 \textrm{Tr}[Y^{\top}\Theta^{-1}(X,X)Y]}{n}}+\textrm{MMD}(S_0,S,\mathcal{H}_{\Theta})$.
>
>     As a result, Eq (19) can be extended to
>
>     $\sqrt{\frac{2}{n}\frac{\textrm{Tr}^2[Y^{\top}Y]}{\mathcal{A}(X,Y)}} + \textrm{MMD}(S_0,S,\mathcal{H}_{\Theta}) \le  \mathcal{B}' \le $
>
>     $\sqrt{\frac{2}{n}\frac{\lambda_{\max}}{\lambda_{\min}}\frac{\textrm{Tr}^2[Y^{\top}Y]}{\mathcal{A}(X,Y)}} + \textrm{MMD}(S_0,S,\mathcal{H}_{\Theta})$.
>
> * In our work, we analyze the generalization bound first and then discuss the relationship between the generalization bound and the discrepancy induced by our active learning method, which keeps the analysis of generalization succinct and leads to an important conclusion (Proposition 1) for our method.
>
>
> 2. Some notations can be simplified to make the formulas cleaner.
> * Thank you for the suggestion. We have simplified the notation and updated Eqs (12)(13)(14) in the revision.

---

> > ### Comment · Reviewer_GLZg · 2022-08-08
> > **Thanks**
> >
> > Thanks for the responses.

---

### Meta-Review · Area_Chair_1CRu · 2022-08-23

**Recommendation:** Accept
**Confidence:** Certain

**Metareview:**

It is the consensus of the reviewers that this paper makes a worthwhile contribution to active deep learning. The author(s)' idea of optimizing for convergence speed is interesting and of potential significance. The meta-reviewer would recommend acceptance of the paper as a poster.

**Award:**

No

---

### Decision · Program_Chairs · 2022-09-14

Accept